# Cooperative Retrieval-Augmented Generation for Question Answering: Mutual Information Exchange and Ranking by Contrasting Layers

**Youmin Ko[1], Sungjong Seo[1], Hyunjoon Kim[1,2*]**
[1]Department of Artificial Intelligence, Hanyang University
[2]Department of Data Science, Hanyang University
{youminkk0213, chrisseo2002, hyunjoonkim}@hanyang.ac.kr

## Abstract

Since large language models (LLMs) have a tendency to generate factually inaccurate output, retrieval-augmented generation (RAG) has gained significant attention as a key means to mitigate this downside of harnessing only LLMs. However, existing RAG methods for simple and multi-hop question answering (QA) are still prone to incorrect retrievals and hallucinations. To address these limitations, we propose CoopRAG, a novel RAG framework for the QA task in which a retriever and an LLM work cooperatively with each other by exchanging informative knowledge, and the earlier and later layers of the retriever model work cooperatively with each other to accurately rank the retrieved documents relevant to a given query. In this framework, we (i) unroll a question into sub-questions and a reasoning chain in which uncertain positions are masked, (ii) retrieve the documents relevant to the question augmented with the sub-questions and the reasoning chain, (iii) rerank the documents by contrasting layers of the retriever, and (iv) reconstruct the reasoning chain by filling the masked positions via the LLM. Our experiments demonstrate that CoopRAG consistently outperforms state-of-the-art QA methods on three multi-hop QA datasets as well as a simple QA dataset in terms of both the retrieval and QA performances. Our code is available.[2]

## 1 Introduction

Since large language models (LLMs) have a tendency to generate factually inaccurate output, retrieval-augmented generation (RAG) has gained significant attention as a key means to mitigate this downside of harnessing only LLMs in various tasks such as knowledge base question answering (KBQA) [4, 21, 24, 79, 87], multi-hop question answering (QA) [19, 20, 42, 86], knowledge graph completion [39, 81, 90], and recommender systems [2, 10, 71, 77]. Recent studies have made various attempts to address the downside of LLMs: (i) applying the fine-granular late interaction scoring mechanism [32, 57] using multi-vector representations, (ii) augmenting a query with its hypothetical answers or query-related concepts by employing LLMs [34, 43, 45], and (iii) allowing LLMs to either generate summaries or a knowledge graph to connect groups of disparate but related passages, and exploring multiple documents from the core concepts of the query in a structure-augmented manner [12, 19, 20].

However, these approaches are still prone to incorrect retrievals and hallucinations [5, 23, 26, 27, 80] especially in simple and multi-hop QA. We hypothesize that these limitations stem from the following three reasons.

---

*Corresponding author.
[2]https://github.com/meaningful96/CoopRAG

First, questions are often too short and limited in information to elicit both the documents most relevant to those questions from the retrieval modules, and high-quality reasoning results from LLMs. Despite the potential of query rewriting methods that form given questions into longer and more helpful queries, their rewriting processes rely heavily on external resources and their supervisions [14, 47], and they do not verify the rewritten queries [52, 88], thus being insufficient to fully draw out the internal knowledge of LLMs.

Second, although the exact causes of incorrect retrievals in RAG are not fully understood, one contributing factor may be the contrastive learning objective, which optimizes dense retrievers by pulling the representations of a query and its relevant documents closer in a shared vector space while simultaneously pushing the query apart from irrelevant documents [72, 73]. This objective can lead to mass-seeking behavior, causing the retriever to retrieve passages that match superficial patterns in the input query rather than passages that contain critical hints or precise factual answers [54, 65]. Empirical studies have shown that retrievers trained with in-batch negatives or heuristic positives on finite data tend to rely on shallow lexical or semantic similarity, rather than retrieving documents that accurately reflect the knowledge encoded in the corpus [65]. From a representational perspective, transformer-based retrievers have been observed to capture more syntactic or surface-level information in lower layers, while higher layers may encode more abstract semantic relations [9, 63].

Third, existing methods fall short in providing LLMs an opportunity to compensate for missing or unconfident knowledge critical to answering the question. Although several existing reasoning strategies [46, 69, 75, 83, 89] enable LLMs to refine their initial outputs, there is a lack of literature on an effective way to complete gaps in knowledge in which LLMs are uncertain or lack sufficient information for the final answer.

To address these issues, we propose a novel RAG framework called cooperative RAG for simple and multi-hop question answering in which a retriever and an LLM work cooperatively with each other by exchanging informative knowledge, and the earlier and later layers of the retriever model work cooperatively with each other to accurately retrieve the most relevant documents to a given query. In this framework, we (i) unroll a question into multiple sub-questions and a masked reasoning chain, (ii) retrieve the documents relevant to this unrolled question, (iii) rerank the documents by contrasting layers of the retriever, and (iv) reconstruct the reasoning chain by filling masked entities via the LLM. Our experiments demonstrate that CoopRAG consistently outperforms state-of-the-art QA methods on three multi-hop QA datasets HotpotQA [82], 2WikiMultihopQA [22], and MuSiQue [66] as well as a single-hop QA dataset NaturalQuestions [36] in terms of both the retrieval and QA performances. Our retrieval method achieves up to 5.3% improvement on the multi-hop QA datasets and up to 35.2% improvement on the single-hop QA dataset over the current state-of-the-art methods [20]. CoopRAG using Gemma2-9B outperforms even prior GPT-4o-mini-based method. Consequently, CoopRAG can create a bidirectional synergy between a retriever and an LLM, i.e., we effectively draw out the internal knowledge of an LLM which will encourage the retriever to provide the documents highly relevant to the query, and the reranking stage effectively harmonizes the internal knowledge of the retriever, and the retrieval results in turn facilitate the confident reasoning of the LLM, thereby enabling accurate retrieval and reasoning for complex questions.

## 2  Related Work

**Query-Augmentation for RAG**. When questions lack sufficient information, LLMs and retrievers are prone to hallucination and inaccurate retrieval, respectively. Various query augmentation methods have emerged to improve document retrieval performance. Several methods [14, 47, 70, 88] extract keywords from the question to have an LLM generate higher-level concepts, hypothetical answers, pseudo-documents, or sub-questions to augment the original question. In [45, 52], LLMs paraphrase a question for use in retrieval. In contrast, Baleen [33] retrieves documents using the original question, summarizes these documents, refines the question by combining it with the summaries, and then performs a second retrieval. However, the augmented queries in these approaches may include incorrect information due to the hallucinations of LLMs, which consequently impairs retrieval accuracy and reasoning performance.

**Dense Retrieval with Pre-trained Language Models**. Dense retrieval has emerged to address the limitations of term-frequency based methods [37, 55, 56] in capturing semantic relationships by encoding queries and documents as dense vectors and computing their similarity. Queries and

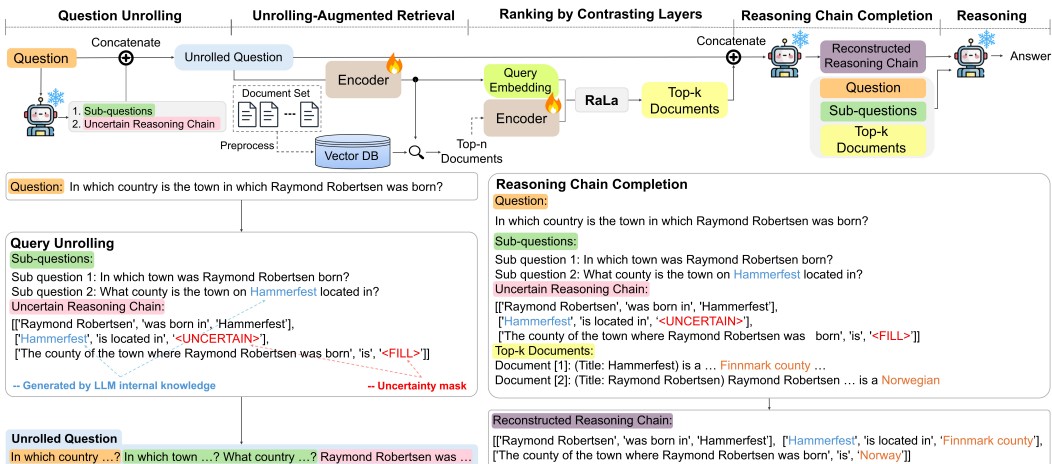

Figure 1: Overview of CoopRAG, which consists of: (i) Question Unrolling, (ii) Unrolling-Augmented Retrieval, (iii) Ranking by Contrasting Layers (RaLa), (iv) Reasoning Chain Completion, and (v) Reasoning

documents are embedded into single vectors to enable efficient similarity computation in [15, 25, 31, 50, 78]. Since such compression introduces representational bottlenecks, ColBERT [32, 57] adopts token-level embeddings. Nevertheless, these methods could not capture distinct types of knowledge encoded across different Transformer layers, leading to distorted similarity scores and thus suboptimal retrieval performance.

**Structure-Augmented RAG**. Retrieval methods that exploit relationships between documents and the structural properties of knowledge have gained attention. RAPTOR [58] and Proposition [6] divide documents into proposition-level segments, and recursively embed, cluster, and summarize them to construct hierarchical representations that capture long-range contextual information. SiReRAG [86] and HopRAG [42] paraphrase complex queries using LLMs, and leverage the paraphrased queries to explore logical connections between document chunks within a knowledge graph (KG). GraphRAG [12] and LightRAG [18] leverage LLMs to extract triples from text, and hierarchically construct KGs to maximize semantic connectivity. HippoRAG [19] and HippoRAG2 [20] build KGs by representing noun phrases as nodes and their relations as edges. However, these methods incur high construction costs, and often produce overly dense graphs with numerous irrelevant reasoning paths.

**Reasoning-Enhanced Approaches for Complex QA**. Reasoning explicitly across multiple steps has been shown to be beneficial for LLMs to solve complex queries through linear reasoning steps [75], multiple branching paths [83], a graph structure [1]. CoK [40] prompts LLMs to generate intermediate knowledge, and utilizes external models to validate that knowledge. In CoQ [89] and question decomposition [53], a question is split into answerable sub-questions to deduct a final answer. However, these approaches do not leverage retrieval-augmentation to validate knowledge in which LLMs have low confidence. IRCoT [67] is a model-agnostic multi-step retrieval framework that interleaves each reasoning step with a document retrieval for mutual enhancement, and can be easily integrated into our method.

## 3 Method

### 3.1 Overview

In this section, we describe a novel RAG framework for question answering. First, we fine-tune a pretrained encoder for retrieval and reranking (Section 3.6). Next, for every document, we obtain the [CLS] token embedding produced by the fine-tuned encoder, and store this in a vector DB in the preprocessing stage.

Figure 1 illustrates the overview of our framework in inference, which consists of five stages: (1) in the question unrolling stage, an LLM leverages its internal knowledge to decompose a question into

multiple sub-questions and a reasoning chain in which uncertain positions are masked (Section 3.2); (2) the question is augmented with the sub-questions and the reasoning chain, a retriever provides top-$n$ documents relevant to the augmented query in unrolling-augmented retrieval (Section 3.3); (3) we rerank the retrieved documents to obtain top-$k$ ($k < n$) documents by contrasting layers in the retriever model (Section 3.4); (4) the LLM reconstructs the reasoning chain by filling the masked positions with the factual evidence in the top-$k$ documents (Section 3.5); (5) the LLM takes the reconstructed reasoning chain, the question, the sub-questions, and the top-$k$ documents as input, and generates the answer for the question.

## 3.2 Question Unrolling

A question often contains limited information to accurately guide both the relevant document retrieval and the correct reasoning of the LLM. In the question unrolling stage, the LLM takes the input question $Q$ as input, and decomposes $Q$ into (i) a set $\mathcal{S} = [s_1, s_2, s_3, ..., s_{|\mathcal{S}|}]$ of sub-questions (ii) an uncertain reasoning chain, i.e., a sequence of triples with masked entities. Inspired by reasoning chains [13, 69], we allow LLM to generate evidence triples that support the step-by-step thinking and the final answer, but the LLM may have low confidence in generating entities of some triples. If so, generating such entities and augmenting the original question with these entities could rather lead to wrong retrievals or hallucinations in reasoning. Therefore, we guide the LLM to mask out these entities instead of generating them. Finally, the sub-questions and the uncertain reasoning chain are concatenated with the original question to foam an unrolled question $U$ as follows:

$$
\begin{aligned}
U &= Q||\mathcal{S}||\mathcal{R} \\
\mathcal{S} &= \{s_1, s_2, s_3, ...s_{|\mathcal{S}|}\} \\
\mathcal{R} &= \{(e_1, r_1, e_1'), (e_2, r_2, e_2'), \ldots, (e_{|\mathcal{R}|}, r_t, \langle\texttt{FILL}\rangle)\}
\end{aligned}
\tag{1}
$$

where $||$ denotes concatenation, and $e_i, r_i, e_i'$ represent the head entity, the relation, and the tail entity, respectively, of the $i$-th triple in the reasoning chain. Each entity stands for either an uncertainty mask $\langle\texttt{UNCERTAIN}\rangle$ if the LLM lacks confidence; a question-related concept otherwise. The tail entity of the last triple in the reasoning chain is designated as the placeholder $\langle\texttt{FILL}\rangle$, which will be substituted with the answer to the original question in the final reasoning step of LLM.

From the following stages, the LLM and the retriever can leverage only internal knowledge about which the LLM is certain. We conduct experiments to validate the effectiveness of question unrolling, which will be discussed in Section 4.6. It is worth mentioning that unlike question unrolling, [53] decomposes a question into sub-questions, and iteratively calls the LLM to modify these sub-questions, and CoQ [89] generates sub-questions, and answers the sub-questions until the LLM elicits the final answer. The analysis and prompt for question unrolling are provided in Appendix E.1-E.4 and Appendix H.1, respectively.

## 3.3 Unrolling-Augmented Retrieval

In the unrolling-augmented retrieval (UAR) stage, we retrieve top-$n$ documents most relevant to the unrolled question from all documents. For this, the fine-tuned encoder computes token-level embeddings of the unrolled question $U$, and the document embedding of every document $D$:

$$
\begin{aligned}
\mathbf{q}_0, \mathbf{q}_1, \ldots, \mathbf{q}_{|U|} &:= \text{Normalize}\big(\text{Encoder}([\texttt{CLS}]\ q_1\ q_2\ \ldots\ q_{|U|})\big) \\
\mathbf{d}_0, \mathbf{d}_1, \ldots, \mathbf{d}_{|D|} &:= \text{Normalize}\big(\text{Encoder}([\texttt{CLS}]\ d_1\ d_2\ \ldots\ d_{|D|})\big)
\end{aligned}
\tag{2}
$$

where Normalize stands for $L_2$ normalization, $q_1, \ldots, q_{|U|}$ represent tokens in the unrolled question $U$, and $d_1, \ldots, d_{|D|}$ represent tokens in document $D$. Let $\mathbf{q}_i$ and $\mathbf{d}_i$ denote the embedding of the $i$-th token in $U$ and $D$ respectively. The [CLS] token embeddings of $U$ and $D$ are represented by $\mathbf{q}_0$ and $\mathbf{d}_0$, respectively.

Next, we retrieve a set $\mathcal{D}_U = \{D_1, D_2, ..., D_n\}$ of top-$n$ documents most relevant to $U$ based on the cosine similarity between $\mathbf{q}_0$ and $\mathbf{d}_0$ for each document. This similarity-based search is performed efficiently by Faiss [30].

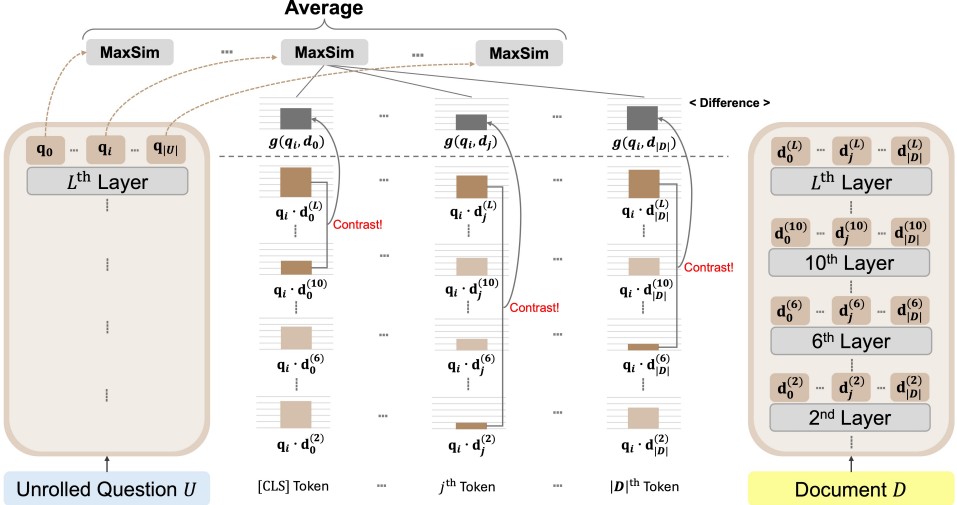

Figure 2: Illustration of the Ranking by Contrasting Layers (RaLa) process before optimization, corresponding to Equation 3.

### 3.4 Ranking by Contrasting Layers

Motivated by retrieve-rerank-generate pipelines [11, 16, 85], we rerank the top-$n$ documents in $\mathcal{D}_U$ to retain top-$k$ ($k < n$) documents in this reranking stage. Factual knowledge has generally been shown to be localized to particular layers of the Transformer model [7, 8, 48], which in turn can provide an opportunity to take advantage of the multiple layers to tackle the inaccurate retrieval of Transformer-based encoders.

Inspired by DoLa [7], we propose to rerank the top-$k$ documents based on a novel ranking method called ranking by contrasting layers (RaLa). As shown in Figure 2, RaLa leverages the differences in information representation between the lower and higher layers of Transformer encoder. The lower layers focus on syntactic or surface-level information, while higher layers capture more abstract semantic relations. RaLa assigns higher weights to documents with greater similarity differences between these two layers relative to the question. While existing retrievers utilize only top-layer embeddings, RaLa enables document retrieval based on semantic relevance through multi-layered representations.

We divide the entire layers into two to four buckets [7], and randomly select one layer from each bucket. Let a set $\mathcal{C} = \{1, 2, \ldots, |\mathcal{C}|\}$ of candidate layers be the set of the selected middle layers, and $L$ represents the last layer of the encoder. For each document $D \in \mathcal{D}_U$ and each layer $l \in \mathcal{C}$, the hidden states, i.e., early exit, of the $l$-th layer are represented as $\mathbf{d}_0^{(l)}, \mathbf{d}_1^{(l)}, \ldots, \mathbf{d}_{|D|}^{(l)}$.

In the reranking stage, we rerank every document $D \in \mathcal{D}_U$ regarding the unrolled question $U$ by computing the score between them. Every query embedding of $U$ interacts with all document embeddings of $D$ via a MaxSim operator, which computes maximum similarity [32, 57], and the outputs of these operators are averaged across all query tokens. Here, we define the similarity between contextualized embeddings of $q_i$ and $d_j$ as the maximum score gap $g(q_i, d_j)$ between the last layer and the premature layer:

$$\text{score}(U, D) = \text{avg}_{i=0}^{|U|}\big(\max_{j\in\{0,1,\ldots,|D|\}} g(q_i, d_j)\big), \quad \text{where } g(q_i, d_j) = \max_{l\in\mathcal{C}}\big(\langle\mathbf{q}_i, \mathbf{d}_j^{(L)}\rangle - \langle\mathbf{q}_i, \mathbf{d}_j^{(l)}\rangle\big),$$
(3)

$\text{avg}(\cdot)$ denotes the average operation, and $\langle\cdot,\cdot\rangle$ represents the cosine similarity between two vectors. However, the cosine similarity applies to every candidate layer to compute the maximum, which may incur substantial computational overhead. Therefore, in practice, we replace the above score with the average maximum similarity in [32, 57], which is now multiplied by the importance of how relevant the document is to the query by applying the dynamic layer selection:

$$\text{score}_o(U, D) = \omega_{U,D} \cdot \text{avg}_{i=0}^{|U|}\big(\max_{j \in \{0,\ldots,|D|\}} \langle \mathbf{q}_i, \mathbf{d}_j^{(L)} \rangle \big) \tag{4}$$

where the gap-aware weight $\omega_{U,D}$ is defined as $g(q_0, d_0)$ in which $q_0$ and $d_0$ represent the $[\texttt{CLS}]$ token in $U$ and $D$ respectively. The comparative performance analysis between Equations (3) and (4) will be presented in Section 4.6. Further comparisons between different scoring strategies are presented in Appendix E.5.

The motivation for selecting the layer with the highest gap is to ensure that the model would significantly change its output after the selected candidate layer, and thus have a higher chance to include more query-relevant knowledge that does not exist in the early layers before it, which is detailed by the case study in Appendix G.1.

### 3.5 Reasoning Chain Completion and Reasoning

Sometimes, the retriever does not provide sufficient information necessary for the LLM to compensate for missing knowledge, in which the LLM has low confidence during step-by-step reasoning. We claim that the retriever providing this knowledge to LLM can contribute to alleviating hallucinations of the LLM.

In the reasoning chain completion stage, the LLM updates the reasoning chain obtained by query unrolling based on the top-$k$ documents. Given the unrolled question $U$ and the top-$k$ documents, the LLM fills in the uncertainty masks $\langle \texttt{UNCERTAIN} \rangle$ and the final entity $\langle \texttt{FILL} \rangle$ in the reasoning chain based on clues in the documents. Subsequently, the LLM generates the final answer, given the input question, the sub-questions, the reconstructed reasoning chain, and the top-$k$ documents. The comparison of different LLM invocation strategies is presented in Appendix E.6. The prompts for reasoning chain completion and reasoning are provided in Appendices H.2 and H.3, respectively.

### 3.6 Difficulty-Aware Training of Retriever

Training a model primarily on easy examples may lead to overfitting to common or superficial patterns, and harder questions usually correspond to ambiguous, rare, or edge-case scenarios [35, 44, 59]. To tackle this, we fine-tune the pretrained encoder for retrieval through sample-wise loss reweighting that gives more attention to difficult questions which may result in wrong document retrievals.

In finetuning, we unroll each question in the training set, and then we associate that unrolled question with a positive document and a hard negative document by following ColBERT [32]. With a mini-batch $\mathcal{B}$ of size $b$, let $\mathcal{U} = \{U_1, U_2, \ldots, U_b\}$ be a set of the unrolled questions in a mini-batch, and let $\mathcal{D} = \{D_1, D_2, \ldots, D_{2b}\}$ be a set of all documents associated with the unrolled questions, where $D_i$ and $D_{b+i}$ denote the positive and hard negative documents for $U_i$ respectively. For each mini-batch, we compute loss below:

$$\mathcal{L}_{\mathcal{B}} = \sum_{U_i \in \mathcal{U}} \alpha_{U_i} \mathcal{L}(U_i, D_i) \quad \text{where} \quad \mathcal{L}(U_i, D_i) = -\log \frac{\exp(\text{score}_o(U_i, D_i)/\tau)}{\sum_{D \in \mathcal{D}} \exp(\text{score}_o(U_i, D)/\tau)} \tag{5}$$

where the weight $\alpha_{U_i}$ stands for the difficulty of the unrolled question $U_i$, ensuring that queries with higher difficulty receive larger penalties. InfoNCE loss $\mathcal{L}(U_i, D_i)$ for $U_i$, where $\tau$ is a hyperparameter that controls the importance of negative samples, and score is defined in Equation 4.

In our experiments for multi-hop question answering without question unrolling, we observe that the retrieval performance deteriorates as the number of ground-truth (GT) documents relevant to a question increases, and the number of the sub-questions is proportional to the number of GT documents, which indicates that the LLM tends to decompose a complex multi-hop question into more sub-questions. Hence, we set the weight $\alpha_{U_i}$ proportional to the number of sub-questions in $U_i$:

$$\alpha_{U_i} = \log(1 + |\mathcal{S}_{U_i}|) \tag{6}$$

where $\mathcal{S}_{U_i}$ represents a set of sub-questions in $U_i$. In this way, we adjust the importance of sample-wise loss, thereby encouraging the encoder to learn more from challenging cases, which are often more informative and critical for generalization.

# 4 Experiments

## 4.1 Experimental Setup

Table 1: Dataset statistics (test sets)

| Category | Single-hop QA | Multi-hop QA | | | Factual QA | |
|---|---|---|---|---|---|---|
| | NQ | HotpotQA | MuSiQue | 2Wiki | FreshQA | SimpleQA |
| Questions | 1,000 | 1,000 | 1,000 | 1,000 | 553 | 1,000 |
| Documents | 9,633 | 9,811 | 11,656 | 6,119 | 11,062 | 20,686 |

Table 1 shows the statistics of the widely-used datasets adopted in our experiments: HotpotQA [82], MuSiQue [66], and 2WikiMultihopQA (2Wiki) [22] for multi-hop QA, and NaturalQuestions (NQ) [36] for single-hop QA. The test set for each multi-hop QA dataset contains 1,000 samples drawn from HippoRAG, and that for NaturalQuestions consists of 1,000 samples randomly selected from about 27,000 test instances provided by REAL [74] to ensure a fair comparison with baselines. Futhermore, we evaluate on recent factual QA benchmarks, SimpleQA [76] and FreshQA [68], which reflect more up-to-date and challenging settings. MPNet [60] is employed as an encoder for retrieval and reranking, and Gemma2 (9B, 27B), Llama3.3-70B [17] and GPT-4o-mini [51] are used as LLMs. Notably, Llama3.3-70B denotes the Llama3.3-70B-Instruct model throughout all experiments. Detailed experimental setup is described in Appendix A. Baseline methods are explained in Appendix B. In addition to these single-step retrieval methods, we also include the multi-step retrieval method IRCoT [67] as a baseline. The complete results of multi-step retrieval and QA are provided in Appendices C.3 and C.4, respectively.

## 4.2 Retrieval Performance Before and After Fine-Tuning

Table 2: Retrieval performance comparison across single-step methods. The best and second-best performances are presented in bold and underlined, respectively.

| Models | Multi-hop QA | | | | | | Single-hop QA | |
|---|---|---|---|---|---|---|---|---|
| | HotpotQA | | MuSiQue | | 2Wiki | | NQ | |
| | R@2 | R@5 | R@2 | R@5 | R@2 | R@5 | R@2 | R@5 |
| HippoRAG2 (Llama3.3-70B) | 83.5 | 96.3 | 56.1 | 74.7 | 76.2 | 90.4 | 45.6 | 78.0 |
| HippoRAG2 (GPT-4o-mini) | 80.5 | 95.7 | 53.5 | 74.2 | 74.6 | 90.2 | 44.4 | 76.4 |
| SiReRAG (GPT-4o-mini) | 80.0 | 94.8 | 52.5 | 64.9 | 60.6 | 67.6 | 42.3 | 72.5 |
| HopRAG (GPT-4o-mini) | 81.1 | 96.0 | 53.7 | 66.8 | 61.7 | 70.1 | 43.9 | 74.4 |
| CoopRAG (Gemma2-9B) | 87.9 | 95.6 | _59.4_ | _75.5_ | 80.1 | _96.7_ | 71.6 | 88.9 |
| CoopRAG (Gemma2-27B) | _88.3_ | _96.6_ | _59.4_ | **75.7** | **80.8** | **97.2** | 72.8 | 89.5 |
| CoopRAG (Llama3.3-70B) | 86.9 | _96.6_ | 58.2 | 75.3 | _80.6_ | 96.3 | _77.2_ | _90.8_ |
| CoopRAG (GPT-4o-mini) | **88.8** | **96.8** | **59.6** | **75.7** | 80.4 | 96.6 | **80.8** | **92.1** |

Table 2 compares the performance of single-step retrieval for ours and the latest baselines. The complete results for all baselines compared with ours are found in Appendix C.1. CoopRAG (GPT-4o-mini) outperforms all of the competitors across every benchmark. On HotpotQA, it achieves the highest Recall@2 and Recall@5. On NaturalQuestions, it improves upon HippoRAG2 by 35.2% in Recall@2 and 14.1% in Recall@5. These results show that unrolling-augmented retrieval based on the masked reasoning paths benefits both multi-hop and single-hop questions. CoopRAG (Gemma2-9B) even surpasses HippoRAG2 with the much larger LLM, i.e., Llama3.3-70B, demonstrating its effectiveness under limited computational resources. We conduct an analysis demonstrating retrieval efficiency in Appendix D.

## 4.3 QA Performance

Table 3 presents single-step QA performance. CoopRAG (GPT-4o-mini) attains state-of-the-art results on most datasets. Even with Gemma2-9B, it outperforms all baselines, and achieves at least 15.2% higher EM on NaturalQuestions. Reranking by contrasting layers and completing the masked

Table 3: QA performance comparison across single-step methods. The best and second-best performances are denoted in bold and underlined, respectively.

| Models | Multi-hop QA | | | | | | Single-hop QA | |
| | HotpotQA | | MuSiQue | | 2Wiki | | NQ | |
| | EM | F1 | EM | F1 | EM | F1 | EM | F1 |
|---|---|---|---|---|---|---|---|---|
| HippoRAG2 (Llama3.3-70B) | 62.7 | 75.5 | 37.2 | 48.6 | 65.0 | 71.0 | 48.6 | 63.3 |
| HippoRAG2 (GPT-4o-mini) | 56.3 | 71.1 | 35.0 | 49.3 | 60.5 | 69.7 | 43.4 | 60.0 |
| SiReRAG (GPT-4o-mini) | 61.7 | 76.5 | 40.5 | 53.1 | 59.6 | 67.9 | 42.4 | 58.7 |
| HopRAG (GPT-4o-mini) | 62.0 | 76.1 | 42.2 | 54.9 | 61.1 | 68.3 | 42.9 | 59.2 |
| CoopRAG (Gemma2-9B) | 64.4 | 78.1 | 52.2 | 65.2 | 70.0 | 78.1 | 63.8 | 72.7 |
| CoopRAG (Gemma2-27B) | 64.9 | **79.5** | **52.8** | 66.7 | 71.7 | 79.0 | 67.3 | 75.5 |
| CoopRAG (Llama3.3-70B) | 64.7 | 79.0 | 52.6 | 66.6 | 71.2 | 78.8 | 70.9 | 80.3 |
| CoopRAG (GPT-4o-mini) | **65.6** | 78.9 | 52.3 | **67.1** | **71.7** | **79.2** | **72.0** | **82.3** |

Table 4: QA Performance comparison on SimpleQA and FreshQA datasets. The best and second-best performances are denoted in bold and underlined, respectively.

| Methods | SimpleQA | | FreshQA | | | | |
| | EM | F1 | EM | F1 | Correct (↑) | Incorrect (↓) | Not Attempted (↓) |
|---|---|---|---|---|---|---|---|
| HippoRAG2 | 48.2 | 55.0 | 21.3 | 29.5 | 225 | 297 | 31 |
| HopRAG | 50.2 | 58.2 | 21.1 | 28.7 | 233 | 275 | 45 |
| CoopRAG | **58.3** | **67.6** | **26.6** | **35.3** | **283** | **250** | **20** |

reasoning chain based on the documents retrieved by UAR enhance the QA accuracy. More detailed analysis on QA performance is included in Appendix C.2.

We also evaluate CoopRAG on two recent factual QA benchmarks, SimpleQA and FreshQA, where it achieves 16.1% and 26.1% higher EM than HopRAG, respectively. Since FreshQA includes many sentence-level and open-form answers, we follow the ChatGPT grader in SimpleQA [76], which labels predictions as Correct if the prediction fully contains the ground truth without contradiction, Incorrect if any contradiction is present, and Not Attempted if the necessary information is missing but not contradicted. Under this metric, CoopRAG produces 283 correct answers, while HopRAG achieves 233 and HippoRAG2 achieves 225, demonstrating clear improvements in factual correctness.

## 4.4 Impact of Gap-Aware and Difficulty-Aware Weights

Table 5: Ablation study on the effect of difficulty-aware weights ($\alpha_U$ and $\omega_{U,D}$), showing retrieval performance with and without each weight.

| Category | HotpotQA | | MuSiQue | | 2Wiki | |
| | R@2 | R@5 | R@2 | R@5 | R@2 | R@5 |
|---|---|---|---|---|---|---|
| *w/o $\alpha_U,\omega_{U,D}$* | 84.4 | 94.3 | 56.3 | 72.7 | 76.1 | 92.3 |
| *w/o $\alpha_U$* | 87.8 | 95.2 | 58.2 | 74.6 | 79.9 | 96.0 |
| *w/o $\omega_{U,D}$* | 85.6 | 94.5 | 57.0 | 73.8 | 77.7 | 93.5 |
| CoopRAG | **88.1** | **95.9** | **59.6** | **75.7** | **81.4** | **96.6** |

We conduct an analysis to evaluate the impact of the gap- and difficulty-aware weights. Table 5 shows retrieval performance with and without each weight. Removing both $\alpha_U$ and $\omega_{U,D}$ results in a marked decline in performance. Recall@2 falls from 81.4% to 76.1% in 2WikiMultihopQA. Between the two weights, removing $\omega_{U,D}$ produces a larger performance drop. On MuSiQue, dropping $\omega_{U,D}$ reduces Recall@2 from 59.6% to 57.0%, whereas dropping $\alpha_U$ leads to a smaller decrease from 59.6% to 58.2%. These results indicate that reweighting the query-document similarity via the gap-aware weight, which contrasts the hierarchical knowledge of LMs, plays a more significant role in retrieving correct documents. Nevertheless, removing $\alpha_U$ also degrades performance across all datasets. This demonstrates that the number of sub-questions reflecting the question difficulty contributes to retrieval performance gains. We further examine alternative weighting methods in Appendix E.7.

## 4.5 Similarity Score Comparison based on Gap-Aware Weight in RaLa

Table 6: The comparison of the similarity score differences to validate the effectiveness of $\omega_{U,D}$, showing a relative increase of 32.16 in (pos) - (rand. neg.).

| Weight | Score Difference | HotpotQA | MuSiQue | 2Wiki |
|---|---|---|---|---|
| $w/o\ \omega_{U,D}$ | (pos) – (rand. neg.) | 0.2951 | 0.2109 | 0.2882 |
| | (pos) – (distractor) | 0.2161 | 0.1952 | 0.2008 |
| | (distractor) – (rand. neg.) | 0.0790 | 0.0157 | 0.0874 |
| $w/\ \omega_{U,D}$ | (pos) – (rand. neg.) | 0.3900 | 0.3847 | 0.4004 |
| | (pos) – (distractor) | 0.2388 | 0.2139 | 0.2306 |
| | (distractor) – (rand. neg.) | 0.1512 | 0.1708 | 0.1698 |

We conduct an experiment comparing similarity scores to validate the effectiveness of the gap-aware weight in distinguishing positive documents from their distractor and random negative (RN) documents. Distractor documents are harder to distinguish from positive documents than RN documents. For our frameworks with and without the gap-aware weight, Table 6 shows three types of differences: (pos) - (rand. neg.) is the difference in the average similarity between question-positive document pairs and question-RN document pairs; (pos) - (distractor) is that between question-positive document pairs and question-distractor document pairs; (distractor) - (rand. neg.) is that between question-distractor document pairs and question-RN document pairs. Applying $\omega_{U,D}$ increases all score differences on all datasets, indicating enhanced ability to distinguish between positive and negative documents. The relative increase in (pos) - (rand. neg.) is 32.16% on HotpotQA, 82.41% on MuSiQue, and 38.93% on 2WikiMultihopQA. The difference (distractor) - (rand. neg.), the relative difficulty of distinguishing distractors from RNs versus positives, substantially increases by 91.39% on HotpotQA, 987.9% MuSiQue, and 94.28% on 2WikiMultihopQA. These results demonstrate that RaLa effectively separates truly semantically relevant documents from their distractors, and distractors from RNs, which is difficult without RaLa due to locally similar keywords in all of these documents.

## 4.6 Impact of Uncertainty Mask

Table 7: Impact of the uncertainty mask on the retrieval and QA performances (Recall@2 and EM)

| Category | HotpotQA | | MuSiQue | | 2Wiki | |
|---|---|---|---|---|---|---|
| | Retrieval (R@2) | QA (EM) | Retrieval (R@2) | QA (EM) | Retrieval (R@2) | QA (EM) |
| $w/o\ \langle\texttt{UNCERTAIN}\rangle$ | 86.9 | 60.9 | 55.2 | 47.2 | 73.8 | 65.6 |
| $w/\ \langle\texttt{UNCERTAIN}\rangle$ | **88.1** | **65.6** | **59.6** | **52.3** | **80.4** | **71.7** |

Table 8: Entropy decrease ratio based on the uncertainty mask

| Category | HotpotQA | MuSiQue | 2Wiki |
|---|---|---|---|
| $\langle\texttt{FILL}\rangle$ Generation $w/$ & $w/o\ \langle\texttt{UNCERTAIN}\rangle$ | 5.13 | 6.32 | 5.62 |
| Before & After $\langle\texttt{UNCERTAIN}\rangle$ Generation | 8.59 | 8.88 | 9.92 |

We analyze the effect of the uncertainty masks by comparing "w/ $\langle\texttt{UNCERTAIN}\rangle$" (CoopRAG) and "w/o $\langle\texttt{UNCERTAIN}\rangle$", i.e, the variant that does not generate uncertainty masks in question unrolling and does not perform reasoning chain completion, on the retrieval and QA performances. Table 7 presents their Recall@2 and EM. w/ $\langle\texttt{UNCERTAIN}\rangle$ improves retrieval performance by 1.4% on HotpotQA, 8.0% on MuSiQue, and 8.9% on 2WikiMultihopQA. QA performance exhibits similar trends, with EM accuracy increasing across all datasets. These findings demonstrate that the uncertainty masks effectively mitigates errors in uncertain information generated during LLM reasoning, implying that masking out low-confidence tokens prevents negative effects on subsequent inference steps.

In Table 8, we measure how many times the average entropy decreases at the moment of generating $\langle\texttt{FILL}\rangle$ token when using $\langle\texttt{UNCERTAIN}\rangle$ compared to w/o $\langle\texttt{UNCERTAIN}\rangle$, and how many times the average entropy decreases right after $\langle\texttt{UNCERTAIN}\rangle$ generation compared to the moment of generating

$\langle$UNCERTAIN$\rangle$. All datasets exhibit substantial entropy differences, with MuSiQue showing a 6.32-fold decrease in generating $\langle$FILL$\rangle$ by using uncertain masks, and 2WikiMultihopQA decreasing by a factor of 9.92 right after generating uncertain masks. Explicitly marking uncertainty may prevent hallucinations, thus enhancing document retrieval and subsequent reasoning.

### 4.7 Comparison of Retrieval Performance

Table 9: Single-step retrieval performance comparison of different retrievers on HotpotQA, MuSiQue, and 2WikiMultihopQA, evaluated with and without fine-tuning. The best and second-best performances are denoted in bold and underlined, respectively.

| Methods | HotpotQA | | MuSiQue | | 2WikiMultihopQA | |
|---|---|---|---|---|---|---|
| | R@2 | R@5 | R@2 | R@5 | R@2 | R@5 |
| *Without Fine-tuning* | | | | | | |
| Contriever | 57.3 | 74.8 | 32.4 | 43.5 | 55.3 | 65.3 |
| ColBERTv2 | 64.7 | 79.3 | 37.9 | 49.2 | 59.2 | 68.2 |
| ReSCORE | 68.2 | 80.9 | 38.6 | 49.1 | 62.2 | 73.3 |
| RaLa | **73.9** | **86.1** | **45.9** | **60.7** | **64.6** | **75.0** |
| *With Fine-tuning* | | | | | | |
| Contriever | 63.4 | 80.5 | 36.8 | 44.9 | 60.6 | 68.2 |
| ColBERTv2 | 78.2 | 89.3 | 52.6 | 68.5 | 71.6 | 82.1 |
| ReSCORE | 82.3 | 92.6 | 54.9 | 72.3 | 78.8 | 95.1 |
| RaLa | **88.8** | **96.8** | **59.6** | **75.7** | **80.4** | **96.6** |

Table 9 compares the retrieval performance of RaLa and recent baselines before and after fine-tuning. Even without fine-tuning, RaLa consistently outperforms Contriever, ColBERTv2, and ReSCORE across all datasets. On HotpotQA, for example, RaLa achieves an R@2 of 73.9%, clearly higher than 68.2% of ReSCORE, with similar advantages observed on MuSiQue and 2WikiMultihopQA.

After fine-tuning, the performance gap becomes more evident. RaLa reaches 88.8% R@2 and 96.8% R@5 on HotpotQA, yielding 7.9 and 4.5% improvements over ReSCORE, respectively. On MuSiQue and 2WikiMultihopQA, RaLa also consistently outperforms ReSCORE, confirming its superior retrieval capability and adaptability across diverse QA benchmarks.

## 5 Conclusion and Limitation

We propose a novel QA framework, where a retriever and an LLM cooperatively exchange mutually useful context, and multiple layers of the retriever jointly contribute to accurate document reranking. Question unrolling allows the LLM to identify positions in which hallucinations are likely to occur, UAR enables LLM to deliver the useful query required to compensate for its uncertain knowledge to the retriever, contrastive reranking enables the retriever to provide appropriate documents that can boost the confidence of LLM, and LLM can confidently answer the question via reasoning chain completion. Extensive experiments on single-hop and multi-hop QA benchmarks demonstrate that CoopRAG achieves state-of-the-art performance in retrieval accuracy and QA quality.

Despite our achievements, pretrained LMs cannot embed long documents due to their sequence length constraints, and like ColBERT, our dense retrieval method might incur rising computational costs for MaxSim operations as the number of tokens grows. Further validation on KBQA [61, 84] and domain-specific datasets [3, 28, 29], and extending our framework beyond passage-based retrieval to knowledge graph QA remains a promising direction for future research.

## Acknowledgments and Disclosure of Funding

This work was supported by the Institute of Information & Communications Technology Planning & Evaluation (IITP) grant funded by the Korea government (MSIT) (No. RS-2020-II201373, Artificial Intelligence Graduate School Program (Hanyang University)); and by the Basic Science Research Program through the National Research Foundation of Korea (NRF) funded by the Ministry of Education (No. RS-2025-25424679).

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

# Appendices

In this supplementary material, we provide further details on the following topics:

- **Appendix A: Implementation Details**
- **Appendix B: Baselines**
- **Appendix C: Overall Performance**

  - C.1: Single-step Retrieval Performance
  - C.2: Single-step QA Performance
  - C.3: Multi-step Retrieval Performance
  - C.4: Multi-step QA Performance

- **Appendix D: Efficiency**

  - D.1: Retrieval Efficiency
  - D.2: Reasoning Efficiency

- **Appendix E: Analysis of CoopRAG**

  - E.1: Impact of Question Unrolling with Varying Similarities
  - E.2: Impact of the Number of Sub-questions
  - E.3: Impact of the Length of the Reasoning Chain
  - E.4: Performance Comparison of Different Question Unrolling Methods
  - E.5: Impact of Contrastive Reranking Strategies
  - E.6: Impact of Separating Reasoning Steps on Model Performance
  - E.7: Impact of Alternative Weighting Methods Based on Sub-Questions and Reasoning Chains
  - E.8: Comparison of Retrieval Performance by Loss Function
  - E.9: Complexity and Latency Analysis
  - E.10: Scalability Analysis

- **Appendix F: Hyperparameter Sensitivity**

  - F.1: Impact of Mini-batch Size
  - F.2: Impact of Temperature in InfoNCE Loss
  - F.3: Impact of Bucket Size

- **Appendix G: Case Study**

  - G.1: Example of RaLa
  - G.2: Example of End-to-End Process
  - G.3: Example of Retrieval Error
  - G.4: Example of Reasoning Error

- **Appendix H: Prompt**

  - H.1: Question Unrolling
  - H.2: Reasoning Chain Completion
  - H.3: QA Reasoning
  - H.4: Multi-step Key Extraction

# A Implementation Details

In our weighted InfoNCE loss, the temperature parameter $\tau$ is set to 0.05. We select the best performing batch size of 40. The maximum sequence length is configured as 512, and the bucket size is set to 3. For in-batch training, we select one random negative sample and one distractor for each question. Details of the hyperparameter sensitivity experiments are provided in Appendix F.

We employ MPNet [60] as an encoder, and Gemma2 (9B, 27B)[62], Llama3.3-70B[17], and GPT-4o-mini [51] as LLMs. Training is performed on two NVIDIA A6000 GPUs, and inference on a single A6000 GPU. For every multi-hop QA dataset, the encoder is trained for 5 epochs, requiring about 5 hours for HotpotQA, 2 hours for MuSiQue, and 8 hours for 2WikiMultihopQA (2Wiki). For NaturalQuestions (NQ), the encoder is trained for 8 epochs, taking approximately 2 hours. Following previous work [19, 20], retrieval performance is assessed using Recall@2 and Recall@5, while QA performance is evaluated with Exact Match (EM) and F1-score.

# B Baselines

We adopt three types of comparative approaches: (i) The classic retrievers **BM25** [55], **Contriever** [25], **ColBERTv2** [57], **Proposition** [6], and **GTR** [50]; (ii) Large embedding models which perform on the BERT leaderboard [64], including **GTE-Qwen2-7B-Instruct** [41], **GritLM-7B** [49], and **NV-Embed-v2** [38]; (iii) Structure-augmented RAG approaches, including **RAPTOR** [58], **HippoRAG** [19], **HippoRAG2** [19], **SiReRAG** [86], and **HopRAG** [42].

# C Overall Performance

## C.1 Single-step Retrieval Performance

Table 10: Overall retrieval performance of single-step methods. The best and second-best performances are presented in bold and underlined, respectively. We use different random seeds for each run, and conduct five runs to report the mean with a maximum standard deviation of ±0.3.

| Models | Multi-hop QA | | | | | | Single-hop QA | |
| --- | --- | --- | --- | --- | --- | --- | --- | --- |
| | HotpotQA | | MuSiQue | | 2Wiki | | NQ | |
| | R@2 | R@5 | R@2 | R@5 | R@2 | R@5 | R@2 | R@5 |
| *Simple Baselines* | | | | | | | | |
| BM25 | 57.3 | 74.8 | 32.4 | 43.5 | 55.3 | 65.3 | 28.2 | 56.1 |
| Contriever | 58.4 | 75.3 | 34.8 | 46.6 | 46.6 | 57.5 | 29.1 | 54.6 |
| GTR (T5-base) | 59.3 | 73.9 | 37.4 | 49.1 | 60.2 | 67.9 | 35.0 | 63.4 |
| Proposition (ColBERTv2) | 63.9 | 78.1 | 37.8 | 50.1 | 55.9 | 64.9 | 33.1 | 62.2 |
| ColBERTv2 | 64.7 | 79.3 | 37.9 | 49.2 | 59.2 | 68.2 | 36.8 | 64.3 |
| *Large Language Models* | | | | | | | | |
| GTE-Qwen2-7B-Instruct | 75.8 | 89.1 | 48.1 | 63.6 | 66.7 | 74.8 | 44.7 | 74.3 |
| GritLM-7B | 79.2 | 92.4 | 49.7 | 65.9 | 67.3 | 76.0 | 46.2 | 76.6 |
| NV-Embed-v2 (7B) | 84.1 | 94.5 | 52.7 | 69.7 | 67.1 | 76.5 | 45.3 | 75.4 |
| *Structure-augmented RAG* | | | | | | | | |
| RAPTOR (Llama3.3-70B) | 76.8 | 86.9 | 47.0 | 57.8 | 58.3 | 66.2 | 40.3 | 68.3 |
| RAPTOR (GPT-4o-mini) | 78.6 | 90.2 | 49.1 | 61.0 | 58.4 | 66.0 | 40.5 | 69.4 |
| HippoRAG (Llama3.3-70B) | 60.4 | 77.3 | 41.2 | 53.2 | 71.9 | 90.4 | 21.3 | 44.4 |
| HippoRAG (GPT-4o-mini) | 60.1 | 78.5 | 41.8 | 52.4 | 68.4 | 87.0 | 21.6 | 45.1 |
| HippoRAG2 (Llama3.3-70B) | 83.5 | 96.3 | 56.1 | 74.7 | 76.2 | 90.4 | 45.6 | 78.0 |
| HippoRAG2 (GPT-4o-mini) | 80.5 | 95.7 | 53.5 | 74.2 | 74.6 | 90.2 | 44.4 | 76.4 |
| SiReRAG (GPT-4o-mini) | 80.0 | 94.8 | 52.5 | 64.9 | 60.6 | 67.6 | 42.3 | 72.5 |
| HopRAG (GPT-4o-mini) | 81.1 | 96.0 | 53.7 | 66.8 | 61.7 | 70.1 | 43.9 | 74.4 |
| CoopRAG (Gemma2-9B) | 87.9 | 95.6 | 59.4 | 75.5 | 80.1 | 96.7 | 71.6 | 88.9 |
| CoopRAG (Gemma2-27B) | 88.3 | 96.6 | 59.4 | **75.7** | **80.8** | **97.2** | 72.8 | 89.5 |
| CoopRAG (Llama3.3-70B) | 86.9 | 96.6 | 58.2 | 75.3 | 80.6 | 96.3 | 77.2 | 90.8 |
| CoopRAG (GPT-4o-mini) | **88.8** | **96.8** | **59.6** | **75.7** | 80.4 | 96.6 | **80.8** | **92.1** |

We compare the performance of single-step retrieval across several benchmark datasets. Table 10 shows that structure-augmented RAG consistently outperforms simple baselines, and achieves the best results on all datasets. CoopRAG (GPT-4o-mini) records the highest scores on every benchmark except 2WikiMultihop, in which

CoopRAG (Gemma2-27B) leads. On NaturalQuestions, CoopRAG (GPT-4o-mini) improves Recall@2 by 34.6% and Recall@5 by 15.5% over the previous state-of-the-art, GritLM-7B. Although NaturalQuestions requires only one document for inference, over 71% of questions in this dataset are decomposed into two or more sub-questions, indicating that question unrolling benefits both single-hop and multi-hop questions. Our methods with different LLMs, i.e., Gemma2-9B, Gemma2-27B, and Llama3.3-70B, achieve superior performance across all datasets, and CoopRAG (Gemma2-9B) excels despite its smaller number of parameters, demonstrating the effectiveness of our method in resource-constrained settings.

## C.2  Single-step QA Performance

Table 11: Overall QA performance comparison across single-step methods. The best and second-best performances are presented in bold and underlined, respectively.

| Models | Multi-hop QA | | | | | | Single-hop QA | |
|---|---|---|---|---|---|---|---|---|
| | HotpotQA | | MuSiQue | | 2Wiki | | NQ | |
| | EM | F1 | EM | F1 | EM | F1 | EM | F1 |
| *Llama3.3-70B* | | | | | | | | |
| BM25 | 52.0 | 63.4 | 20.3 | 28.8 | 47.9 | 51.2 | 44.7 | 59.0 |
| Contriever | 51.3 | 62.3 | 24.0 | 31.3 | 38.1 | 41.9 | 45.0 | 58.9 |
| GTR (T5-base) | 50.6 | 62.8 | 25.8 | 34.6 | 49.2 | 52.8 | 45.5 | 59.9 |
| GTE-Qwen2-7B-Instruct | 58.6 | 71.0 | 30.6 | 40.9 | 55.1 | 60.0 | 46.6 | 62.0 |
| GritLM-7B | 60.7 | 73.3 | 33.6 | 44.8 | 55.8 | 60.6 | 46.8 | 61.3 |
| NV-Embed-v2 (7B) | 62.8 | 75.3 | 34.7 | 45.7 | 57.5 | 61.5 | 47.3 | 61.9 |
| RAPTOR | 56.8 | 69.5 | 20.7 | 28.9 | 47.3 | 52.1 | 36.9 | 50.7 |
| GraphRAG | 55.2 | 68.6 | 27.3 | 38.5 | 51.4 | 58.6 | 30.8 | 46.9 |
| LightRAG | 2.0 | 2.4 | 0.5 | 1.6 | 9.4 | 11.6 | 8.6 | 16.6 |
| HippoRAG | 52.6 | 63.5 | 26.2 | 35.1 | 65.0 | 71.8 | 43.0 | 55.3 |
| HippoRAG2 | 62.7 | 75.5 | 37.2 | 48.6 | 65.0 | 71.0 | 48.6 | 63.3 |
| *GPT-4o-mini* | | | | | | | | |
| RAPTOR | 50.6 | 64.7 | 27.7 | 39.2 | 39.7 | 48.4 | 37.8 | 54.5 |
| GraphRAG | 51.4 | 67.6 | 27.0 | 42.0 | 45.7 | 61.0 | 38.0 | 55.5 |
| LightRAG | 9.9 | 20.2 | 2.0 | 9.3 | 2.5 | 12.1 | 2.8 | 15.4 |
| HippoRAG | 46.3 | 60.0 | 24.0 | 35.9 | 59.4 | 67.3 | 37.2 | 55.2 |
| HippoRAG2 | 56.3 | 71.1 | 35.0 | 49.3 | 60.5 | 69.7 | 43.4 | 60.0 |
| SiReRAG | 61.7 | 76.5 | 40.5 | 53.1 | 59.6 | 67.9 | 42.4 | 58.7 |
| HopRAG | 62.0 | 76.1 | 42.2 | 54.9 | 61.1 | 68.3 | 42.9 | 59.2 |
| CoopRAG (Gemma2-9B) | 64.4 | 78.1 | 52.2 | 65.2 | 70.0 | 78.1 | 63.8 | 72.7 |
| CoopRAG (Gemma2-27B) | 64.9 | **79.5** | **52.8** | 66.7 | **71.7** | 79.0 | 67.3 | 75.5 |
| CoopRAG (Llama3.3-70B) | 64.7 | 79.0 | 52.6 | 66.6 | 71.2 | 78.8 | 70.9 | 80.3 |
| CoopRAG (GPT-4o-mini) | **65.6** | 78.9 | 52.3 | **67.1** | **71.7** | **79.2** | **72.0** | **82.3** |

We compare the performance of single-step QA across all the benchmark datasets. As shown in Table 3, CoopRAG (GPT-4o-mini) achieves state-of-the-art performance on most datasets. In particular, when using Gemma2-9B, CoopRAG achieves 15.2% higher EM on NaturalQuestions than the previous state-of-the-art method. These results demonstrate that reranking by contrasting layers and completing the masked reasoning chain based on the documents retrieved by unrolling-augmented retrieval enhance the QA accuracy.

## C.3  Multi-step Retrieval Performance

IRCoT [67] is a representative framework for multi-step retrieval and QA. However, it employs a simple structure in which the LLM performs chain-of-thought reasoning over the question and candidate documents, and augments the question until it judges the problem solved. Moreover, its basic prompting design requires a lot of examples, i.e., 13–15 shots. To overcome these limitations, we propose the KeyExtract method: the LLM first evaluates whether LLM can infer the answer from the initially retrieved candidate documents; if LLM cannot, LLM extracts a key sentence from those documents, and appends it to the unrolled question for iterative re-retrieval. KeyExtract operates effectively with only three shots, and can be applied even to small LLMs such as Gemma2-9B.

To demonstrate the effectiveness of the KeyExtract method, we compare multi-step retrieval performance across various methods and datasets. Table 12 presents the results against baselines. Our approach outperformed existing methods by a wide margin on all datasets. In particular, KeyExtract + CoopRAG (GPT-4o-mini) achieves a 34.6% improvement in Recall@2 and a 17.1% improvement in Recall@5 on HotpotQA compared to IRCoT +

Table 12: Overall retrieval performance comparison across multi-step methods. The best and second-best performances are denoted in bold and underlined, respectively. We used different random seeds for each run and, for each dataset and evaluation metric, conducted five runs to report the mean, with a maximum standard deviation of ±0.21.

| Models | HotpotQA | | MuSiQue | | 2Wiki | |
|---|---|---|---|---|---|---|
| | R@2 | R@5 | R@2 | R@5 | R@2 | R@5 |
| *Simple Baselines* | | | | | | |
| IRCoT + BM25 (Default) | 65.6 | 79.0 | 34.2 | 44.7 | 61.2 | 75.6 |
| IRCoT + Contriever | 65.9 | 81.6 | 39.1 | 52.2 | 51.6 | 63.8 |
| IRCoT + ColBERTv2 | 67.9 | 82.0 | 41.7 | 53.7 | 64.1 | 74.4 |
| *Structure-augmented RAG* | | | | | | |
| IRCoT + HippoRAG (Contriever) | 65.8 | 82.3 | 43.9 | 56.6 | 75.3 | 93.4 |
| IRCoT + HippoRAG (ColBERTv2) | 67.0 | 83.0 | 45.3 | 57.6 | 75.8 | 93.9 |
| IRCoT + CoopRAG (GPT-4o-mini) | 87.3 | 90.9 | 62.2 | 74.9 | 81.0 | 95.9 |
| KeyExtract + CoopRAG (Gemma2-9B) | 88.6 | 93.9 | 62.9 | 76.8 | 81.8 | 97.2 |
| KeyExtract + CoopRAG (GPT-4o-mini) | **90.2** | **97.2** | **64.5** | **78.6** | **83.6** | **97.6** |

HippoRAG (ColBERTv2), the latest structure-aware RAG model. On MuSiQue, our method improves Recall@2 and Recall@5 by 42.3 % and 36.5%, respectively, and on 2WikiMultihopQA by 10.3% and 3.9%, respectively. Notably, applying KeyExtract yields an average performance gain of 3.31% over the IRCoT approach. These findings demonstrate that KeyExtract retrieves relevant documents more effectively for questions requiring complex reasoning, even with fewer examples. The prompt for KeyExtract is in Appendix H.4

## C.4  Multi-step QA Performance

Table 13: Overall QA performance comparison across multi-step methods. The best and second-best performances are denoted in bold and underlined, respectively.

| Models | Reader (LLM) | HotpotQA | | MuSiQue | | 2Wiki | |
|---|---|---|---|---|---|---|---|
| | | EM | F1 | EM | F1 | EM | F1 |
| IRCoT + ColBERTv2 | GPT-4o-mini | 45.5 | 58.4 | 19.1 | 30.5 | 35.4 | 45.1 |
| IRCoT + HippoRAG | GPT-3.5-turbo | 45.7 | 59.2 | 21.9 | 33.3 | 47.7 | 62.7 |
| IRCoT + CoopRAG | GPT-4o-mini | 64.3 | 75.5 | 52.3 | 65.9 | **72.2** | 77.7 |
| KeyExtract + CoopRAG | Gemma2-9B | 64.9 | 77.9 | 52.5 | 66.1 | 71.5 | **78.9** |
| KeyExtract + CoopRAG | GPT-4o-mini | **66.7** | **79.2** | **53.8** | **68.6** | 72.2 | 78.2 |

We compare the multi-step QA performance. As shown in Table 13, CoopRAG outperforms all existing methods by a substantial margin. We observe that applying IRCoT to CoopRAG using the same baseline setup results in a at least 40.7% improvement in EM on HotpotQA compared to the baselines. Notably, combining KeyExtract with CoopRAG (GPT-4o-mini) yields an additional gain of over 5.3%. This trend holds across the other datasets as well. On MuSiQue, our model achieves an EM score 145.7% higher than the previous state-of-the-art, i.e., IRCoT + HippoRAG. These results demonstrate that CoopRAG is highly compatible to multi-step QA frameworks such as IRCoT.

## D  Efficiency

### D.1  Retrieval Efficiency

Table 14 compares the efficiency of different retrieval methods, evaluating search latency and accuracy (R@2) for CoopRAG versus HippoRAG2. For a fair comparison, all experiments are conducted by using GPT-4o mini. The results show that CoopRAG retrieves faster on HotpotQA and MuSiQue. Although CoopRAG is slightly slower on 2WikiMultihopQA, the difference is negligible. Importantly, despite the matching or exceeding retrieval speed of HippoRAG2, CoopRAG delivers substantially higher accuracy across all datasets. On HotpotQA, we achieve R@2 = 88.1% compared to 80.5% for HippoRAG2 (+7.6%); on MuSiQue, R@2 = 59.6% versus 53.5 (+6.1%); and on 2WikiMultihopQA, R@2 = 80.4% versus 74.6% (+5.8%). These results demonstrate that our approach significantly improves retrieval accuracy without sacrificing the retrieval speed.

Table 14: Comparison of retrieval efficiency between our model and HippoRAG2

| Model | HotpotQA | | MuSiQue | | 2Wiki | |
|---|---|---|---|---|---|---|
| | Time (s) | R@2 | Time (s) | R@2 | Time (s) | R@2 |
| HippoRAG2 | 2.25 | 80.5 | 2.11 | 53.5 | **2.34** | 74.6 |
| CoopRAG | **2.04** | **88.1** | **1.98** | **59.6** | 2.39 | **80.4** |

## D.2 Reasoning Efficiency

Table 15: Comparison of the number of LLM calls per question across preprocessing, retrieval, and reasoning stages for CoopRAG, HippoRAG2, and HopRAG.

| Methods | Retrieval | | QA | | LLM calls per question | | |
|---|---|---|---|---|---|---|---|
| | Recall@2 | Recall@5 | EM | F1 | Preprocessing | Retrieve | Reasoning |
| HippoRAG2 | 80.5 | 95.7 | 56.3 | 71.1 | 4 | 1 | 1 |
| HopRAG | 81.1 | 96.0 | 62.0 | 76.1 | 12 | 14.96 | 1 |
| CoopRAG (unified) | 88.8 | 96.8 | 63.1 | 76.6 | 0 | 1 | 1 |
| CoopRAG | 88.8 | 96.8 | 65.6 | 78.9 | 0 | 1 | 2 |

To demonstrate the reasoning efficiency of CoopRAG, we compare the number of LLM calls required by CoopRAG and state-of-the-art baselines, HippoRAG2 and HopRAG, across preprocessing, retrieval, and inference stages. As shown in Table 15, HippoRAG2 and HopRAG make 4 and 12 calls per question during preprocessing, while CoopRAG requires none. This difference arises because both baselines repeatedly invoke the LLM for each document to generate triples. During retrieval, HopRAG incurs an additional average of 14.9 calls per question due to its graph-based iterative triple extraction method. In the inference stage, CoopRAG makes two calls per question, one for reasoning chain completion and another for reasoning. Despite this, CoopRAG achieves up to 16.5% higher EM compared to the baselines. CoopRAG (Unified), which combines the two reasoning steps into a single call, also outperforms both baselines. In summary, CoopRAG demonstrates clear efficiency and effectiveness, requiring fewer LLM calls while achieving superior performance.

## E Analysis

### E.1 Impact of Question Unrolling with Varying Similarities

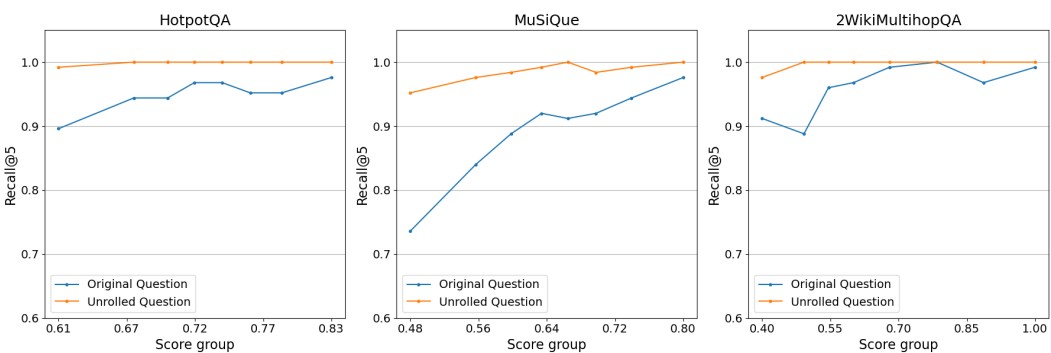

Figure 3: Mean Recall@5 against the score between questions and positive documents. We sort all of the question-positive document pairs by their similarity scores, divide them into equal-sized groups, and measure the average similarity scores and average Recall@5 for each group.

We confirm that question unrolling, which enhances the original question using the internal knowledge of LLM, significantly increases similarity with the positive, i.e., ground-truth, document compared to the original question. Figure 3 shows the difference in retrieval performance according to the similarity between a question and its ground-truth document. The retrieval performance for using the unrolled question consistently outperforms that for using the original question. As the similarity decreases, their performance gap increases. This indicates that when the question is structured and enhanced by effectively utilizing the internal knowledge, the retriever can more accurately retrieve highly relevant documents.

## E.2 Impact of the Number of Sub-questions

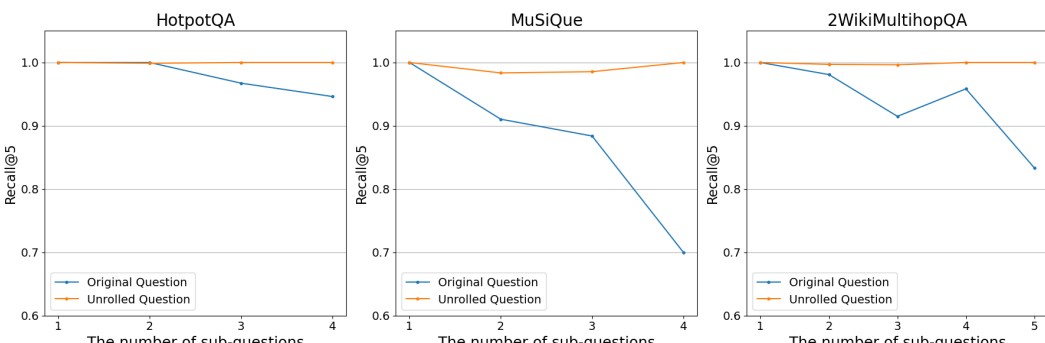

Figure 4: Effectiveness of question unrolling with respect to sub-question complexity.

We investigate the effect of the number of sub-questions on retrieval performance to analyze the effectiveness of question unrolling. Figure 4 presents the how average performance varies with the number of sub-questions, where "Original question" refers to the CoopRAG variant without using question unrolling, and "Unrolled Question" refers to CoopRAG. Across all datasets, CoopRAG consistently outperforms the variant. As the number of sub-questions increases, the performance for the original question declines, whereas that of CoopRAG remains stable. On MuSiQue, the performance gap between CoopRAG and the variant reaches approximately 30%, i.e., the largest observed difference, when there are four sub-questions. This finding demonstrates that question unrolling becomes increasingly beneficial for more complex questions. The results suggest that relying solely on the original question makes it difficult to retrieve appropriate documents for questions requiring complex reasoning, and that our question unrolling method effectively overcomes this limitation.

## E.3 Impact of the Length of Reasoning Chain

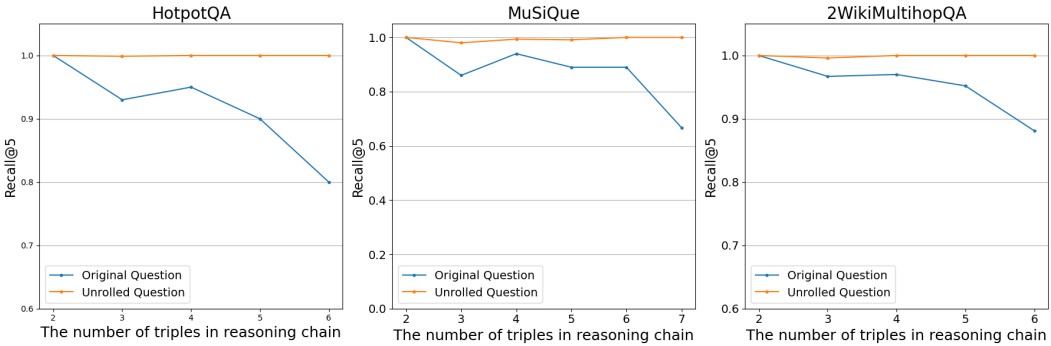

Figure 5: Effectiveness of Question Unrolling with respect to reasoning chain complexity (number of triples).

We conduct an additional experiment to analyze the impact of the length of a reasoning chain on retrieval performance. Figure 5 presents the how average performance varies with the number of triples in the reasoning chain, where "Original question" refers to the CoopRAG variant without using question unrolling, and "Unrolled Question" refers to CoopRAG. Across all datasets, CoopRAG consistently outperforms the variant "Original question". As the number of triples increases, the performance of the variant declines sharply, while the performance of CoopRAG remains stable. On HotpotQA, the performance gap between the original and unrolled questions widens to approximately 20%, when six triples are included in the reasoning chain. On MuSiQue, a similarly large gap of about 35% appears when seven triples are used. These findings indicate that the more complex the reasoning process, the greater the benefit of question unrolling. The results demonstrate that for questions requiring complex reasoning chains, relying solely on the original question makes it difficult to retrieve appropriate documents, demonstrating that question unrolling effectively overcomes this limitation.

Table 16: Ablation study on the effect of question unrolling. The best and second-best performances are denoted in bold and underlined, respectively.

| Category | HotpotQA | | MuSiQue | | 2Wiki | |
|---|---|---|---|---|---|---|
| | R@2 | R@5 | R@2 | R@5 | R@2 | R@5 |
| MQ | 71.6 | 82.8 | 43.7 | 56.5 | 71.5 | 88.6 |
| MQ + SQ | 80.3 | 90.9 | 51.6 | 67.9 | 77.7 | 89.9 |
| MQ + RC | 86.4 | 93.9 | 57.7 | 73.3 | 78.7 | 92.9 |
| MQ + SQ + RC | **88.1** | **95.9** | **59.6** | **75.7** | **80.4** | **96.6** |

## E.4 Performance Comparison of Different Question Unrolling Methods

Table 16 shows retrieval results for different question unrolling techniques: MQ stands for the CoopRAG variant without question unrolling, MQ + SQ stands for the variant that decomposes an input question into only sub-questions, MQ + RC decomposes an input question into only a masked reasoning chain, and MQ + SQ + RC denotes CoopRAG. For HotpotQA, MQ + SQ increases Recall@2 of MQ by 12%. MQ + SQ + RC further increases Recall@2 to 88.1%, with a 23% gain over MQ. On MuSiQue, MQ yields Recall@2 of 43.7% but MQ + SQ + RC raises it to 59.6%, achieving a 36% improvement. Notably, MQ + RC alone moves Recall@2 from 51.6% to 59.6%. A similar pattern appears on 2WikiMultihopQA where MQ yields 71.5% and the full combination reaches 80.4%, resulting in a 12.1% increase. Across all datasets, MQ + SQ improves performance of MQ, and MQ + SQ + RC enhances MQ + SQ further. These findings demonstrate that decomposing the main question into sub questions and explicitly representing the reasoning steps with uncertain reasoning chains improves the document retrieval accuracy for complex questions.

## E.5 Impact of Contrastive Reranking Strategies

Table 17: The effect of different contrastive reranking strategies on retrieval and QA performance within CoopRAG.

| Strategy | HotpotQA | | MuSiQue | | 2Wiki | |
|---|---|---|---|---|---|---|
| | R@2 | R@5 | R@2 | R@5 | R@2 | R@5 |
| Contrasting token embeddings | 86.6 | 94.4 | 52.2 | 69.9 | 77.1 | 91.2 |
| Contrasting similarity scores | **88.4** | **96.6** | **60.0** | **76.2** | **81.8** | **96.7** |
| Optimization based on $\omega_{U,D}$ | 88.1 | 95.9 | 59.6 | 75.7 | 81.4 | 96.6 |

$$\text{score}(U, D) = \text{avg}_{i=0}^{|U|}\big(\max_{j\in\{0,1,\dots,|D|\}}\langle\mathbf{q}_i, \mathbf{d}_j^{(l^*)}\rangle\big) \quad \text{where, } l^* = \arg\max_{1\le l < L}\big\|\mathbf{d}_j^{(L)} - \mathbf{d}_j^{(l)}\big\|_2 \tag{7}$$

Table 17 compares retrieval performance under the RaLa framework using three contrastive reranking strategies: contrasting token embeddings in Equation 7, contrasting similarity scores in Equation 3, and the optimization based on the gate-aware weight $\omega_{U,D}$ in Equation 4. According to the table, contrasting similarity scores, i.e., Equation 3, achieves the best performance, demonstrating that reflecting fine-grained information change through token-level similarity differences is highly effective. In contrast, contrasting token embeddings yields the lowest performance. Although it effectively captures token-wise information differences, it fails to preserve the overall contextual semantics of the document. The contrasted embedding vector converges toward zero if key tokens exhibit minimal change in embedding, offering little contribution to scoring. Ultimately, we adopt the optimization strategy, i.e., Equation 4. Although this strategy shows slightly lower performance than contrasting similarity scores, it offers several advantages. First, relying solely on the [CLS] token allows the encoder to effectively retain the document's global semantics. Second, this strategy more efficient than contrasting similarity scores by reducing the training time by more than a factor of four. Considering both effectiveness and efficiency, the optimization strategy is the most practical choice. In summary, this strategy delivers near-best performance while greatly enhancing computational efficiency.

## E.6 Impact of Separating Reasoning Steps on Model Performance

We compare QA performance between: (1) unifying reasoning chain completion and reasoning steps in a single LLM call, and (2) separating them into independent calls. We evaluate both small open-source LLMs (Gemma2-9B and Gemma2-27B), and larger API-based LLMs (GPT-4o-mini and GPT-o3). Across all LLMs in Table 18, the separated-call approach consistently outperforms the unified-call approach. Notably, the performance drop for the unified-call becomes more pronounced as model size decreases. For the smallest LLM,

Table 18: QA performance under unified vs separate strategies for reasoning chain completion and reasoning for different LLMs.

| LLM-call Strategy | LLM | HotpotQA | |
| --- | --- | --- | --- |
| | | EM | F1 |
| Unified | Gemma2-9B | 58.7 | 72.7 |
| Separated | Gemma2-9B | **64.2** | **77.8** |
| Unified | Gemma2-27B | 59.9 | 75.5 |
| Separated | Gemma2-27B | **64.9** | **79.5** |
| Unified | GPT-4o-mini | 63.1 | 76.6 |
| Separated | GPT-4o-mini | **64.7** | **78.8** |
| Unified | GPT-o3 | 70.2 | **83.3** |
| Separated | GPT-o3 | **71.1** | 82.8 |

i.e., Gemma2-9B, using the unified-call results in over an 6% reduction in both EM and F1 compared to the separated-call. These results suggest that compact LMs incur greater cognitive load when processing complex reasoning in a single step, leading to degraded performance. CoopRAG makes three LLM calls in total, which is the same as HippoRAG2 [20] and fewer than HopRAG [42].

### E.7 Impact of Alternative Weighting Methods Based on Sub-Questions and Reasoning Chains

Table 19: Retrieval performance comparison using the number of sub-questions and the length of a reasoning-chain as weights for CoopRAG.

| Difficulty-Aware Weight | HotpotQA | | MuSiQue | | 2Wiki | |
| --- | --- | --- | --- | --- | --- | --- |
| | R@2 | R@5 | R@2 | R@5 | R@2 | R@5 |
| *w/o* difficulty-aware weight | 87.8 | 95.2 | 58.2 | 74.6 | 79.9 | 96.0 |
| Number of sub-questions | **88.1** | **95.9** | 59.6 | 75.7 | **81.4** | **96.6** |
| Reasoning chain length | 87.8 | 95.8 | **60.3** | **76.4** | **81.4** | 96.5 |

We compare CoopRAG and its two variants: (1) removing the difficulty-aware weight from CoopRAG, i.e., w/o $\alpha_{U_i}$, (2) using the number of sub-questions as a difficulty-aware weight, i.e., CoopRAG, and (3) using the length of a masked reasoning chain as a difficulty-aware weight. Table 19 shows their retrieval performances. Applying these weights during training consistently improves performance across all datasets compared to w/o $\alpha_{U_i}$, indicating that reweighting by the question complexity enhances the retrieval capability of the encoder. Both reweighting approaches achieve similar overall performance, though subtle differences emerge. On HotpotQA and 2WikiMultihopQA, CoopRAG achieves Recall@2 of 88.1% and 81.4%, respectively, exceeding 87.8% and matching 81.4% obtained by applying the reasoning chain length, respectively. Overall, CoopRAG is slightly superior on HotpotQA, whereas reweighting by the reasoning chain length produces higher Recall@2 and Recall@5 on MuSiQue than CoopRAG. As a result, the optimal reweighting strategy may vary with dataset characteristics. Both CoopRAG and the variant using the reasoning chain length reflect the question complexity effectively, as evidenced by their comparable performance.

### E.8 Comparison of Retrieval Performance by Loss Function

Table 20: Comparison of retrieval performance using different loss functions (Recall@2).

| Loss | HotpotQA | MuSiQue | 2Wiki |
| --- | --- | --- | --- |
| Cross-entropy | 77.5 | 30.9 | 63.3 |
| InfoNCE | **88.1** | **59.6** | **80.4** |

We conduct experiments comparing InfoNCE loss adopted by CoopRAG and cross-entropy loss to analyze the impact of various loss functions on retrieval performance. Table 20 shows Recall@2 across the three datasets. The experimental results demonstrate that CoopRAG using InfoNCE loss achieves significantly higher performance than the variant using cross-entropy loss on all datasets. The performance differences are substantial. These findings indicate that contrastive learning such as InfoNCE loss can provide more accurate retrieval results.

### E.9 Complexity and Latency Analysis

Table 21: Latency comparison between HippoRAG2 and RaLa. We report average latency (in seconds) for retrieval, QA, and total end-to-end processing.

| Methods | HotpotQA | | | MuSiQue | | |
|---|---|---|---|---|---|---|
| | Retrieval | QA | Total | Retrieval | QA | Total |
| HippoRAG2 | 2.25 | 1.96 | 4.21 | 2.11 | 1.81 | 3.92 |
| CoopRAG | 2.04 | 1.86 | 3.90 | 1.98 | 1.65 | 3.63 |

We analyze the computational complexity and latency of RaLa in comparison with HippoRAG2 to contextualize the practical trade-offs of our method. RaLa adopts a retriever–reranker pipeline. The time complexity for retrieval is $O(N \cdot d)$, where $N$ is the number of documents and $d$ is the embedding dimension. For reranking, the naive time complexity without our optimization, as described in Equation 3 of Section 3.4, is $O(L_q \cdot L_d \cdot d \cdot |\mathcal{C}|)$, where $L_q$ and $L_d$ denote the query and document lengths, respectively, and $|\mathcal{C}|$ is the bucket size. By applying our optimization strategy in Equation 4, only the [CLS] tokens are compared per bucket, which reduces the complexity to $O(L_q \cdot L_d \cdot d)$ and makes it asymptotically identical to that of ColBERT. All reported results are obtained using this optimized version.

As shown in Table 21, RaLa consistently achieves lower latency than HippoRAG2 across both datasets. On HotpotQA, RaLa reduces retrieval latency from 2.25s to 2.04s and QA latency from 1.96s to 1.86s, leading to an overall reduction in total latency from 4.21s to 3.90s. A similar trend is observed on MuSiQue, where RaLa achieves 1.98s retrieval latency and 1.65s QA latency, compared to 2.11s and 1.81s for HippoRAG2, reducing the total latency from 3.92s to 3.63s. These results show that RaLa not only delivers stronger retrieval and QA performance but also processes queries more efficiently.

### E.10 Scalability Analysis

Table 22: Retrieval performance of HippoRAG2, HopRAG, and CoopRAG when varying the candidate size from 10,000 to 60,000 documents. Performance gain (%) denotes the relative improvement of CoopRAG over the best baseline. The best and second-best performances are denoted in bold and underlined, respectively.

| Methods | Candidate size | | | | | |
|---|---|---|---|---|---|---|
| | 10000 | 20000 | 30000 | 40000 | 50000 | 60000 |
| HippoRAG2 | 95.7 | 86.5 | 78.2 | 56.3 | 41.7 | 33.2 |
| HopRAG | 96.0 | 85.6 | 77.1 | 58.8 | 43.4 | 38.9 |
| CoopRAG | **96.8** | **88.6** | **81.9** | **68.7** | **59.6** | **56.8** |
| Performance Gain (%) | 0.83 | 2.43 | 4.73 | 16.84 | 37.33 | 46.02 |

We evaluate the scalability of CoopRAG by varying the candidate size in Wikipedia from 10,000 to 60,000 documents. As shown in Table 22, CoopRAG achieves consistently higher retrieval performance than HippoRAG2 and HopRAG across all candidate sizes. With 10,000 candidates, CoopRAG shows a marginal gain of 0.83% over the best baseline, but the advantage becomes more substantial as the candidate pool grows. At 40,000 candidates, CoopRAG outperforms baselines by more than 16%, and at 60,000 candidates the gap widens to 46.02%. These results highlight that CoopRAG is less affected by the presence of distracting negatives and preserves strong recall even in large-scale retrieval settings.

## F Hyperparameter Sensitivity

### F.1 Impact of Mini-batch Size

We conduct experiments on HotpotQA, MuSiQue, and 2WikiMultihopQA to evaluate the impact of the mini-batch size on retrieval performance. Figure 6 shows Recall@2 for mini-batch sizes of 30, 40, and 50 across the datasets. A mini-batch size of 40 yields the highest Recall@2, 88.1% on HotpotQA, 59.6% on MuSiQue, and 80.4% on 2WikiMultihopQA. We further observe that performance drops sharply when mini-batch size exceeds 50.

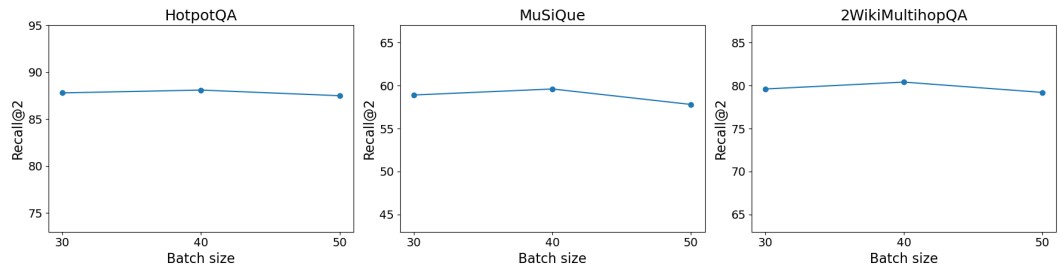

Figure 6: Hyperparameter sensitivity analysis on batch size, showing Recall@2 for batch sizes of 30, 40, and 50 across datasets (HotpotQA, MuSiQue, and 2WikiMultihopQA).

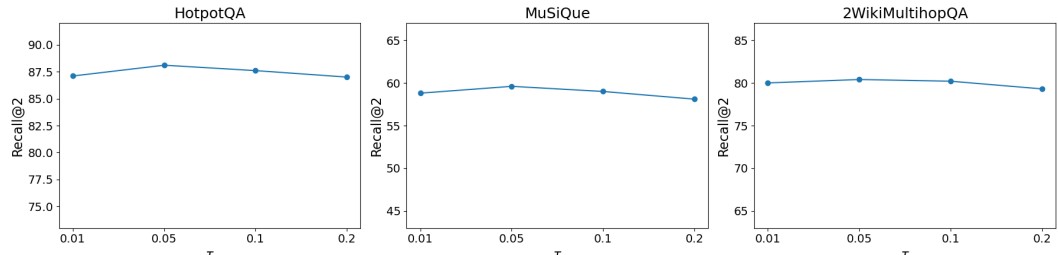

Figure 7: Hyperparameter sensitivity analysis on temperature in InfoNCE loss (Recall@2 for $\tau$ values of 0.01, 0.05, 0.1, and 0.2).

## F.2 Impact of Temperature in InfoNCE Loss

We conduct experiments on HotpotQA, MuSiQue and 2WikiMultihopQA to analyze the effect of the temperature hyper-parameter $\tau$ in InfoNCE loss on retrieval performance. Figure 7 shows Recall@2 for the $\tau$ values of 0.01, 0.05, 0.1 and 0.2 across the datasets. The highest Recall@2 on HotpotQA, MuSiQue and 2WikiMultihopQA is yielded when $\tau = 0.05$. Performance declines sharply once $\tau$ reaches 0.2.

## F.3 Impact of Bucket Size

Table 23: Effect of bucket size on retrieval performance

| Bucket Size | HotpotQA | | MuSiQue | | 2Wiki | |
|---|---|---|---|---|---|---|
| | R@2 | R@5 | R@2 | R@5 | R@2 | R@5 |
| 2 | 87.7 | 94.5 | 58.8 | 75.3 | 80.1 | 95.2 |
| 4 | 88.1 | 95.9 | 59.6 | 75.7 | 80.4 | 96.6 |
| 6 | 88.1 | 96.2 | 59.9 | 75.5 | 80.6 | 97.0 |
| 12 | **88.3** | **96.3** | **60.1** | **75.9** | **81.0** | **97.3** |

We conduct comparative experiments to analyze the impact of bucket size on retrieval performance. MPNet with 12 layers is used as encoder, and the number of layers in each bucket is determined by dividing the total number of layers by the bucket size. Table 23 the experimental results with varying bucket sizes. A bucket size of 12, which utilizes all layers for contrasting, achieves the highest performance across all three datasets. However, there is a trade-off between the retrieval accuracy and the computational efficiency. Raising the bucket size from 4 to 12 yields only a marginal performance improvement, while increasing the training time by 4.6 times. Similarly, increasing the bucket size from 4 to 6 results in negligible accuracy gain but 1.8 times longer training time. To identify an optimal balance between performance and efficiency, we evaluated various bucket size configurations, and ultimately selected a bucket size of 4, which offers near-best performance while keeping computational resource usage at a reasonable level.

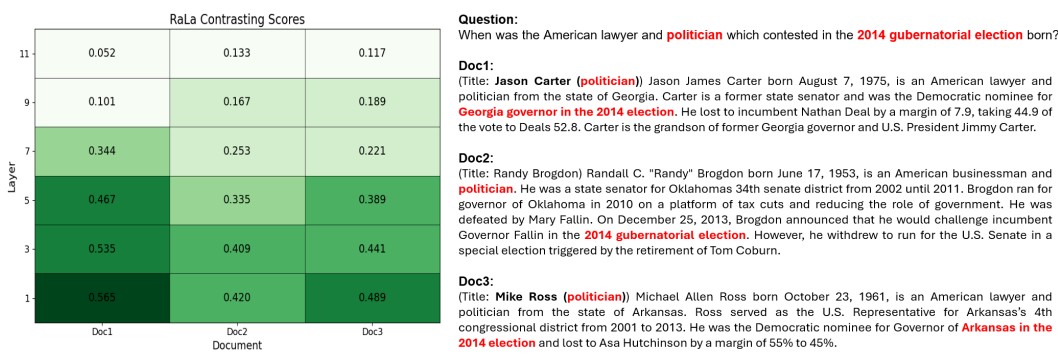

Figure 8: Illustrative example of RaLa. For each document, the heatmap displays the scores obtained by applying RaLa between the final, i.e., 12th, layer and premature odd-numbered layers (1, 3, 5, 7, 9, 11). An input question, a ground-truth document, and two negative documents are described to the right.

# G    Case Study

## G.1    Example of RaLa

We conduct a case study to demonstrate the effect of RaLa. Figure 8 illustrates a heatmap of the score, Equation (4), between the question and the three documents, i.e., Doc1, Doc2, Doc3, to the right, where the x-axis of the heatmap represents document indices, and the y-axis represents the odd-numbered-layers, i.e., 1, 3, 5, 7, 9, and 11. Document 1 is the ground-truth document, whereas Documents 2 and 3 are negative. Note Document 1 provides decisive clues by specifying "Georgia governor in the 2014 election" along with the birthdate August 7, 1975. In contrast, Document 2 contains the 2014 gubernatorial election, but the politician in this document withdrew in the election, and Document 3 shares similar context with the question: "Democratic nominee for Governor of Arkansas in the 2014 election."

We confirm that RaLa effectively highlights representational differences between the question and the documents, enabling clear identification of the correct document. Without RaLa, the average MaxSim scores at the final layer are 0.7666 for Document 1, 0.8110 for Document 2, and 0.7915 for Document 3, indicating higher surface-level similarity for the negative documents. This occurs because the core keywords in the question appear in the negative documents, e.g., "politician" appears in all of the three documents, and "2014 gubernatorial election" appears only in Document 2. In the heatmap of RaLa, score increases are larger in the lower layers, e.g., Document 1 reaches the highest score of 0.565 at layer 1, while the score for layer 11 is 0.052. As a result, RaLa moves beyond surface-level keyword matching by contrasting semantic differences between the final layer and earlier layers, thereby effectively distinguishing the truly relevant document among the three documents.

## G.2    Example of End-to-End Process

Figure 9 illustrates the reasoning chain completion and reasoning stages. It clearly demonstrates how the input and output formats evolve throughout the progression of our approach.

## G.3    Example of Retrieval Error

We compare and evaluate the retrieval performance before and after applying uncertainty masks. We quantitatively analyze changes in accuracy and relevance of the top-$n$ retrieved documents. As previously mentioned in Section 4.6, we observed that hallucinations may occur in the LLM's reasoning chain generation when it produces uncertain entities instead of masking out entities it considers uncertain. For example, as shown in Figure 10, we can see differences in the retrieved documents when the LLM marks uncertain entities with ⟨UNCERTAIN⟩ versus when it does not. This difference in the retrieved documents leads to hallucinations during the final reasoning process, thus encouraging the LLM to produce incorrect answers.

## G.4    Example of Reasoning Error

As shown in Figure 11, we focus on analyzing how uncertainty masks affect the reasoning process. This case illustrates the reasoning process for a question comparing the death dates of film directors. With uncertainty masks, the LLM marks uncertain information, i.e., the directors' death dates, with ⟨UNCERTAIN⟩ tags, and accurately retrieves the relevant documents, and fills in this information during the reasoning chain completion

**Case Study: End-to-End Process**

**Question:** Which film has the director who died later, *45 Calibre Echo* or *Bons Baisers De Hong Kong*?

**SUB_Q1:** Who directed the film *45 Calibre Echo*?
**SUB_Q2:** Who directed the film *Bons Baisers De Hong Kong*?
**SUB_Q3:** What was the date of death for the director of *45 Calibre Echo*?
**SUB_Q4:** What was the date of death for the director of *Bons Baisers De Hong Kong*?

**Uncertain Reasoning Chain:**
```
[["45 Calibre Echo", "was directed by", "Bruce M. Mitchell"],
["⟨UNCERTAIN⟩", "was directed by", "⟨UNCERTAIN⟩"],
["⟨UNCERTAIN⟩", "died on", "⟨UNCERTAIN⟩"],
["Yvan Chiffre", "died on", "⟨UNCERTAIN⟩"],
["Between the directors of the two films", "the one who died later is", "⟨FILL⟩"]]
```

**Top-5 Retrieved Documents:**

- **Document[1]** (Title: 45 Calibre Echo) 45 Calibre Echo is a 1932 American western film directed by Bruce M. Mitchell and starring Jack Perrin, Ben Corbett and Elinor Fair.

- **Document[2]** (Title: Bons Baisers de Hong Kong) Bons Baisers de Hong Kong also known as From Hong Kong with Love is a 1975 French film directed by Yvan Chiffre. It is a parody of James Bond movies featuring Les Charlots with scenes shot in Hong Kong. Mickey Rooney featured in the film as well as Bernard Lee and Lois Maxwell, stars of the James Bond films who appeared as M and Miss Moneypenny, respectively. It was filmed at the Shaw Brothers studios in Hong Kong.

- **Document[3]** (Title: Yvan Chiffre) Yvan Chiffre 3 March 1936 27 September 2016 was a French director, producer, and stunt coordinator. He is the father of Philippe Chiffre, Romain Chiffre and the grandfather of Cesar Chiffre.

- **Document[4]** (Title: Bruce M. Mitchell) Bruce M. Mitchell November 16, 1883 September 26, 1952 was an American film director and writer active during the silent film era from 1914 to 1934. With the advent of sound films in the 1930s, Mitchell abandoned directing and became an actor, appearing mainly in bit roles.

- **Document[5]** (Title: Won in the Clouds) Won in the Clouds is a 1928 American silent film directed by Bruce M. Mitchell and starring Al Wilson. Like many actors in the silent film era, Wilson did not survive the transition to" talkies", with" Won in the Clouds", one of his last films.

**Reconstructed Reasoning Chain:**
```
[["45 Calibre Echo", "was directed by", "Bruce M. Mitchell"],
["Bons Baisers de Hong Kong", "was directed by", "Yvan Chiffre"],
["Bruce M. Mitchell", "died on", "September 26, 1952"],
["Yvan Chiffre", "died on", "27 September 2016"],
["Between the directors of the two films", "the one who died later is", "Yvan
Chiffre"]]
```

**Final Answer:** *Bons Baisers De Hong Kong*
**Ground Truth:** *Bons Baisers De Hong Kong*

Figure 9: Case study of the CoopRAG end-to-end process

stage. This results in the correct answer "Bons Baisers De Hong Kong." In contrast, without uncertainty masks, the LLM generates incorrect death dates for Bruce M. Mitchell as "December 31, 2020" and Yvan Chiffre as "May 15, 2000", thus ultimately producing the wrong answer "Bruce M. Mitchell". This clearly demonstrates how uncertainty masks play a crucial role in preventing hallucinations, and enabling accurate reasoning of LLMs.

**Case Study: Effectiveness of Uncertainty Mask on Retrieval**

**Question:** Who was in charge of the place where Castricum is located?

**SUB_Q1:** What province is Castricum located in?
**SUB_Q2:** What is the capital of the province where Castricum is found?
**SUB_Q3:** Who is the King's Commissioner of North Holland?

**Ground Truth:** *Johan Remkes*
**Number of Golden Docs:** *2*

**<With Uncertainty Mask>**

**Uncertain Reasoning Chain:**
```
[["Castricum", "is located in", "North Holland province"],
["North Holland's capital", "is", "Haarlem"],
["The King's Commissioner of North Holland", "is", "<UNCERTAIN>"],
["The person in charge of the place where Castricum is located", "is", "<FILL>"]]
```

**Top-5 Retrieved Documents:**

- **Document[1]** (Title: Castricum) Castricum is a seaside town in the province of North Holland. It draws in a fair share of tourists who mainly come to visit the beach and nearby dune landscape. [. . . ]
- **Document[2]** (Title: North Holland) The capital and seat of the provincial government is Haarlem, and the provinces largest city is the Netherlands capital Amsterdam. The King's Commissioner of North Holland is Johan Remkes, serving since 2010. [. . . ]
- **Document[3]** (Title: Dutch Republic) In 1579 a number of the northern provinces of the Low Countries signed the Union of Utrecht, in which they promised to support each other in their defence against the Spanish army. [. . . ]
- **Document[4]** (Title: Dutch Republic) During the Republic, any person who wished to hold public office had to conform to the Reformed Church and take an oath to this effect. [. . . ]
- **Document[5]** (Title: Capital of the Netherlands) Although the proper legal status of Amsterdam as capital of the Netherlands is of recent date, the city has been uniformly recognised as capital ever since 1814. [. . . ]

**Final Answer:** *Johan Remkes*

**<Without Uncertainty Mask>**

**Generated Reasoning Chain:**
```
[["Castricum", "is associated with", "castra"],
["Castra", "means", "Roman military camp"],
["Roman military camp", "was ruled by", "Nero Augustus Caesar"],
["The person in charge of the place where Castricum is located", "is", "Nero
Augustus Caesar"]]
```

**Top-5 Retrieved Documents:**

- **Document[1]** (Title: Castricum) Castricum is a seaside town in the province of North Holland. It draws in a fair share of tourists who mainly come to visit the beach and nearby dune landscape. [. . . ]
- **Document[2]** (Title: Forged from the Love of Liberty) Patrick S. Castagne composed the words and music of the National Anthem in 1962. [. . . ]
- **Document[3]** (Title: Last Supper (del Castagno)) The Last Supper 14451450 is a fresco by the Italian Renaissance artist Andrea del Castagno, located in the refectory of the convent of SantApollonia, now the "Museo di Cenacolo di SantApollonia", and accessed through a door on Via Ventisette Aprile at the corner with Santa Reparata, in Florence, region of Tuscany. [. . . ]
- **Document[4]** (Title: Galicia (Spain)) The Roman legions first entered the area under Decimus Junius Brutus in 137136 BC, but the country was only incorporated into the Roman Empire by the [. . . ]
- **Document[5]** (Title: Saint Peter) According to Christian tradition, Peter was crucified in Rome under Emperor Nero Augustus Caesar. [. . . ]

**Final Answer:** *Nero Augustus Caesar*

Figure 10: Case study of the uncertainty masks on retrieval, illustrating differences in retrieved documents when the LLM marks uncertain entities with ⟨UNCERTAIN⟩ versus when it does not.

**Case Study: Effectiveness of Uncertainty Mask on Reasoning**

**Question:** Which film has the director who died later, *45 Calibre Echo* or *Bons Baisers De Hong Kong*?

**SUB_Q1:** Who directed the film *45 Calibre Echo*?
**SUB_Q2:** Who directed the film *Bons Baisers De Hong Kong*?
**SUB_Q3:** What was the date of death for the director of *45 Calibre Echo*?
**SUB_Q4:** What was the date of death for the director of *Bons Baisers De Hong Kong*?

**Ground Truth:** *Bons Baisers De Hong Kong*

**<With Uncertainty Mask>**

**Uncertain Reasoning Chain:**
```
[["45 Calibre Echo", "was directed by", "Bruce M. Mitchell"],
["⟨UNCERTAIN⟩", "was directed by", "⟨UNCERTAIN⟩"],
["⟨UNCERTAIN⟩", "died on", "⟨UNCERTAIN⟩"],
["Yvan Chiffre", "died on", "⟨UNCERTAIN⟩"],
["Between the directors of the two films", "the one who died later is", "⟨FILL⟩"]]
```

**Reconstructed Reasoning Chain:**
```
[["45 Calibre Echo", "was directed by", "Bruce M. Mitchell"],
["Bons Baisers De Hong Kong", "was directed by", "Yvan Chiffre"],
["Bruce M. Mitchell", "died on", "September 26, 1952"],
["Yvan Chiffre", "died on", "27 September 2016"],
["Between the directors of the two films", "the one who died later is", "Yvan Chiffre"]]
```

**Final Answer:** *Bons Baisers De Hong Kong*

**<Without Uncertainty Mask>**

**Generated Reasoning Chain:**
```
[["45 Calibre Echo", "was directed by", "Bruce M. Mitchell"],
["Bons Baisers De Hong Kong", "was directed by", "Yvan Chiffre"],
["Bruce M. Mitchell", "died on", "December 31, 2020"],
["Yvan Chiffre", "died on", "May 15, 2000"],
["Between the directors of the two films", "the one who died later is", "Bruce M. Mitchell"]]
```

**Final Answer:** *Bruce M. Mitchell*

Figure 11: Case study of the uncertainty masks in reasoning

# H Prompt

## H.1 Question Unrolling

---

**Prompt: Question Unrolling**

You are an assistant specialized in multi-hop question answering and logical decomposition. Your task is to analyze complex questions, break them down into reasoning steps, and create structured representations of the reasoning chain.

**Follow these steps exactly when processing a question:**

1. Upon receiving a question, determine the number of hops (reasoning steps) needed to answer it.

- Utilize all information you know to the fullest extent in your reasoning processes.

- Draw on your existing knowledge without external search tools.

2. Provide a brief logical explanation of how the question can be broken down into sequential reasoning steps.

- This explanation should help understand the reasoning path without revealing specific factual answers.

3. Decompose the original question into a series of independent sub-questions that follow the logical reasoning path.

- Each sub-question must be answerable with a single piece of information.

4. Create a structured reasoning chain using triples in the format "[Head, Relation, Tail]".

- Draw specific facts, relationships, and entities from your knowledge to clearly define each component.

- The final triple's tail should always be "$\langle$FILL$\rangle$" to represent the answer to the original question.

**\*\*Important Constraints\*\***
When creating sub-questions:

- You may use expressions from the Original Question.

- Do not use pronouns or placeholders (like "it", "this person", etc.). Always use clear, specific terms and fully spelled-out entity names.

- Each sub-question must be completely self-contained and independently answerable without requiring context from other sub-questions.

- If the Original Question involves comparing two elements, create a separate sub-question that explicitly asks for this comparison using the full names of the entities being compared.

- If you are uncertain about an entity or lack confident knowledge about it, replace the entity with "$\langle$UNCERTAIN$\rangle$"

**\*\*Final Output Format\*\***

**Hop Count:** [number]

**Reasoning Structure:** [brief explanation]

**Sub-questions:** ["Sub-question 1", "Sub-question 2", "Sub-question 3", ...]

**Triple Reasoning Chain:** [["Triple1_head", "Triple1_relation", "Triple1_tail"],
["Triple2_head", "Triple2_relation", "Triple2_tail"],
...
["TripleN_head", "TripleN_relation", "$\langle$FILL$\rangle$"]]

---

Figure 12: Prompt for Question Unrolling.

## H.2 Reasoning Chain Completion

---

**Prompt: Reasoning Chain Completion**

You are a Triple Verification agent designed to precisely complete reasoning chains for Multi-hop Question-Answering. Your task is to examine the provided documents, main question, sub-questions, and reasoning chain containing placeholders marked as "⟨FILL⟩" or "⟨UNCERTAIN⟩".

**Follow these steps exactly when processing a question:**

1. For each placeholder ("⟨FILL⟩" or "⟨UNCERTAIN⟩" or else), strictly replace it with the exact phrase or word explicitly found in the provided documents. Do not paraphrase or introduce synonyms. Ensure every placeholder is replaced using only verbatim text extracted from the documents. If the reasoning chain is incomplete or additional triples are necessary for accurate reasoning, explicitly add new triples by strictly using exact phrases or words found in the provided documents.

2. Your completed reasoning chain must contain exclusively verbatim terms from the documents. Do not include introductory phrases, explanations, or any additional commentary.

3. The final output should strictly follow the triple list format provided below without any deviations:
```
[["Triple1_head", "Triple1_relation", "Triple1_tail"],
["Triple2_head", "Triple2_relation", "Triple2_tail"],
...
["TripleN_head", "TripleN_relation", "⟨FILL⟩"]]
```

**Input Format**

**DOCUMENTS:**
`[context]`

**MAIN_QUESTIONS:**
`[question]`

**SUB_QUESTIONS:**
`[sub_questions]`

**REASONING_CHAIN:**
`[chain]`

**Output Format**

**Reconstructed Reasoning Chain:**
`[Reconstructed reasoning chain formatted exactly as the triple list shown above, with all "⟨FILL⟩" or "⟨UNCERTAIN⟩" placeholders accurately replaced using exact phrases or words from the provided documents, and any necessary new triples explicitly included]`

---

Figure 13: Prompt for Reasoning Chain Completion.

## H.3 Reasoning

---

**Prompt: QA Reasoning**

---

You are a Multi-hop Question-Answering inference agent specialized in generating concise and factual answers based strictly on provided documents and a fully completed reasoning chain.

1. Given the documents, the main question, sub-questions, and the reconstructed reasoning chain, carefully infer the most concise and essential answer to the main question.

2. Your answer must strictly use phrases or words explicitly present either in the completed reasoning chain or directly in the provided documents. Do not use synonyms, paraphrasing, or external knowledge.

3. Verify your inferred answer explicitly using the provided documents to ensure its accuracy and factual correctness before finalizing.

4. Do not generate the answer in full sentence. Return only the most concise and essential term as the answer, avoiding any appended descriptions, subtitles, or variations. If the question is binary (e.g., "yes" or "no"), respond explicitly with "yes" or "no" without any additional explanation.

5. Indicate your answer precisely within the delimiters provided below.

**\*\*Input Format\*\***

**DOCUMENTS:**
`[context]`

**MAIN_QUESTIONS:**
`[question]`

**SUB_QUESTIONS:**
`[sub_questions]`

**REASONING_CHAIN:**
`[chain]`

**\*\*Output Format\*\***

**GENERATED_ANSWER:**
$\langle\langle$ANS$\rangle\rangle$`[Your Answer Here]`$\langle\langle$ANS$\rangle\rangle$

---

Figure 14: Prompt for QA Reasoning.

### H.4 Multi-step Key Extraction

---

**Prompt: Multi-step Key Extraction**

You are an expert reasoning agent for multi-hop question answering, using a Beam Retrieval framework to iteratively select and analyze the most relevant document at each step. Your task is to iteratively select the most helpful document among the provided documents in order to deduce the final answer. At each iteration, you will receive top-10 documents in the following fixed format:

**Document [i]:** (Title: [title]) [text]

**For each iteration, you must follow these steps:**

1. Analyze all 5 documents and select the document that contains the most relevant information.

2. From the selected document, choose a single "Key Sentence" that best contributes to deducing the answer.

3. Output a tuple in the exact format below:

**\*\*Output Format\*\***

**([i], "Key Sentence"). So the answer is: [Final Answer]**

- "[i]" is the index of the selected document.

- "Key Sentence" must be exactly the sentence you consider most informative.

- "[Final Answer]" should be the final deduced answer if it is fully determined; if the final answer cannot be deduced, output False (i.e. So the answer is: False).

- Output format must be "([i], "Key Sentence"). So the answer is: [Final Answer]".

---

Figure 15: Prompt for Multi-step Key Extraction.

