# OpenReview forum: "Cooperative Retrieval-Augmented Generation for Question Answering: Mutual Information Exchange and Ranking by Contrasting Layers"
_NeurIPS.cc/2025/Conference — NeurIPS 2025 poster_

### Official Review · Reviewer_aR4a · 2025-06-20

**Clarity:** 3
**Significance:** 2
**Originality:** 3
**Rating:** 4
**Confidence:** 4

**Summary:**

This paper introduces CoopRAG to address inaccurate retrievals and hallucinations in Retrieval-Augmented Generation (RAG) for simple and multi-hop QA by enabling a retriever and an LLM to cooperate via mutual information exchange and layer-contrastive reranking. CoopRAG first unrolls a question into sub-questions and a masked reasoning chain, then performs Unrolling-Augmented Retrieval, applies Ranking by Contrasting Layers (RaLa) for reranking, and finally completes the reasoning chain via LLM-driven entity filling. Experiments on benchmarks show the effectiveness of the proposed approach.

**Questions:**

See the Weakness.

**Ethical Concerns:**

["NO or VERY MINOR ethics concerns only"]

**Final Justification:**

Thanks for the author's responses.

I would suggest that the authors include the case studies and error analyses in the main body of the paper if space is allowed.

After carefully reading the author's response and other reviews, I raised the Quality and Clarity scores but kept the final rating unchanged.

**Limitations:**

yes

**Quality:**

3

**Strengths And Weaknesses:**

Strengths:
1. The topic of the paper is interesting and the proposed approach is novel.
2. The proposed approach show good performance on benchmark datasets.
3. The paper is easy to follow.


Weaknesses:
1. It would be better if the authors can provide case studies or visualizations to illustrate the effectiveness of the proposed approach.
2. It is suggested to perform an error analysis with failed case so that future studies can know how to improve it.
3. The proposed model is evaluated with four LLMs (i.e., Gemma2-9B, Gemma2-27B, Llama3.3-70B, and GPT-4o-mini). It is recommended to also evaluate the baselines on the same group of LLMs and report the results in Table 2.

---

> ### Author Rebuttal · Authors · 2025-07-29
>
> # Dear Reviewer aR4a,
> Thank you very much for your thoughtful and constructive feedback. We truly appreciate your time and valuable comments. We have carefully addressed each of your suggestions and will incorporate all feedback and new results into the final version. If you find our responses satisfactory, we kindly request that you reconsider the evaluation of our work and, if possible, raise the score above the middle range. Your support would be greatly appreciated. If you have any further questions or suggestions, we would be happy to provide clarification. Thank you again for your consideration.
>
> --------
>
> ### **W1) It would be better if the authors can provide case studies or visualizations to illustrate the effectiveness of the proposed approach.**
> We provide four case studies, which are briefly summarized below and described in more detail in Appendix G.
> - **Example of RaLa in Appendix G.1**: We visualize layer-wise similarity as a heatmap and show that RaLa better distinguishes the correct document by reducing surface-level bias compared to standard scoring.
> - **End-to-end Process in Appendix G.2**: Figure 9 presents a case study of the end-to-end CoopRAG process, showing the prompts used at each stage from question decomposition to final answer generation.
> - **Example of Retrieval Error in Appendix G.3**: A case study demonstrates that using uncertainty masks improves retrieval by focusing on reliable information in the reasoning chain, resulting in more accurate and factual answers.
> - **Example of Reasoning Error in Appendix G.4**: We demonstrate that incorporating uncertainty masks into the reasoning chain enables the model to explicitly mark uncertain spans, which are later resolved using retrieved documents. This process prevents reasoning errors and improves factual accuracy.
>
> ------------
>
> ### **W2) It is suggested to perform an error analysis with failed case so that future studies can know how to improve it.**
>
> | Error Type        | HotpotQA | MuSiQue | 2WikiMultihopQA |
> |-------------------|----------|---------|-----------------|
> | Retrieval Error   | 8.96%    | 24.30%  | 10.70%          |
> | Dataset Error     | 38.30%   | 40.60%  | 39.40%          |
> | Reasoning Error   | 52.74%   | 35.10%  | 49.90%          |
>
> **Table 1.** Error type distribution (%) for HotpotQA, MuSiQue, and 2WikiMultihopQA in CoopRAG.
>
> As shown in Table 1, we categorize errors into three types: retrieval error, dataset error, and reasoning error. Retrieval error refers to cases where some or all key documents are missing from the retrieved set, with MuSiQue showing the highest proportion (24.30%) since it typically requires more supporting documents per question.
>
> Dataset error refers to cases (excluding retrieval errors) where the ground truth does not provide sufficient information to answer the question, or where the answer is expressed differently in the document (e.g., "John Fitzgerald Kennedy" vs. "J. F. Kennedy"). We measured these rates by providing the question, ground truth documents, and answer to GPT-4o-mini, resulting in 38.30% for HotpotQA, 40.60% for MuSiQue, and 39.40% for 2WikiMultihopQA.
>
> All remaining errors are classified as reasoning errors, which mainly arise from the LLM's generative limitations or hallucinations due to long context.
>
> -------
> ### **W3) The proposed model is evaluated with four LLMs (i.e., Gemma2-9B, Gemma2-27B, Llama3.3-70B, and GPT-4o-mini). It is recommended to also evaluate the baselines on the same group of LLMs and report the results in Table 2**
>
> As shown in Appendix C.1 and C.2, CoopRAG consistently outperforms state-of-the-art baselines when evaluated with the same LLMs. On NaturalQuestions, CoopRAG (GPT-4o-mini) improves Recall@2 by 34.6% and Recall@5 by 15.5% over GritLM-7B, and achieves 15.2% higher EM than HippoRAG2 when using Gemma-2-9B. For QA performance on 2WikiMultihopQA, CoopRAG outperforms HippoRAG2 by 9.53% in EM with Llama3.3-70B and by 17.3% with GPT-4o-mini. These results demonstrate CoopRAG’s clear advantage in both retrieval and QA tasks across different LLMs.

---

### Official Review · Reviewer_h3Qx · 2025-07-01

**Clarity:** 3
**Significance:** 2
**Originality:** 2
**Rating:** 4
**Confidence:** 3

**Summary:**

This paper presents CoopRAG, a novel RAG framework in which a language model first decomposes the input question into sub-questions and a masked reasoning chain. A retriever then retrieves and reranks documents based on this unrolled query, and finally, the language model fills in the masked components to generate the answer.
For reranking, the authors propose RaLa, a method that computes the maximum difference of cosine similarity between the query and the final layer and a lower layer. his is based on the hypothesis that higher transformer layers capture semantic similarity, while lower layers primarily encode surface-level or stylistic features. The retriever, based on MPNet, is further trained with difficulty-aware weighting to emphasize challenging examples and prevent overfitting to easy questions.
Experimental results across several QA benchmarks, including HotpotQA, 2WikiMultihopQA, and Natural Questions, demonstrate the effectiveness of CoopRAG in both retrieval accuracy and final answer quality.

**Questions:**

- Are other baselines also training their retrieval systems on the target dataset?

**Ethical Concerns:**

["NO or VERY MINOR ethics concerns only"]

**Final Justification:**

I appreciated the authors' response, and the new results addressed my concerns about evaluations (baselines and additional datasets). That being said, I think the technical novelties are somewhat limited, and thus my final score is 4.

**Limitations:**

The authors included a limitation section, such as long-context capabilities and further evaluations on KBQA.

**Quality:**

2

**Strengths And Weaknesses:**

**Strengths**

- A new training methodology for retrieval systems based on previous findings about transformers' internal representation differences: To my knowledge, I am not aware of retrieval training methods that use the same approach (contrasting the last and lower layers).

- Strong results on short-form QA datasets: The experimental results show that the proposed methods outperform HippoRAG2, HopRAG, and SiReRAG.

**Weaknesses**

- Technical novelty: The proposed framework (CoopRAG) seems like a combination of multiple incremental ideas—decomposing/enhancing queries using LLMs before retrieving documents to improve retrieval results has been widely studied in RAG. While I think RaLa is an interesting new idea, the evaluation focuses on comparisons with other recent RAG systems that do not use trained retrievers, making me wonder whether RaLa is indeed better than other retrieval training objectives.

- **Baselines**: Related to the point above, to my knowledge (correct me if I’m wrong), all baseline methods do not train their retrieval systems on the target datasets by default, as their main focus is more on inference-time algorithms and interactions between the retriever and the LM. In contrast, the proposed method trains retrievers on the target datasets. This makes me wonder whether the authors could compare model performance with methods that also train retrieval models on the target datasets.

- **Evaluations**: Evaluations are primarily conducted on simple, short-form QA datasets (synthetic multi-hop and simple single-hop questions). These datasets are often more easily decomposed, so I was curious how the proposed "unrolling" strategies would perform on more natural, complex questions, such as those in SimpleQuestions or natural search queries (e.g., Search Arena queries). Given that intermediate reasoning chains assume knowledge-triple formats, I am afraid this method may not generalize well beyond entity-centric, factoid questions.

---

> ### Author Rebuttal · Authors · 2025-07-29
>
> # Dear Reviewer h3Qx,
> Thank you very much for your thoughtful and constructive feedback. We truly appreciate your time and valuable comments. We have carefully addressed each of your suggestions and will incorporate all feedback and new results into the final version. If you find our responses satisfactory, we kindly request that you reconsider the evaluation of our work and, if possible, raise the score above the middle range. Your support would be greatly appreciated. If you have any further questions or suggestions, we would be happy to provide clarification. Thank you again for your consideration.
>
> ----------
>
> ### **W1) Technical novelty: The proposed framework (CoopRAG) seems like a combination of multiple incremental ideas—decomposing/enhancing queries using LLMs before retrieving documents to improve retrieval results has been widely studied in RAG. While I think RaLa is an interesting new idea, the evaluation focuses on comparisons with other recent RAG systems that do not use trained retrievers, making me wonder whether RaLa is indeed better than other retrieval training objectives.**
>
> Many prior studies have explored decomposing or enhancing queries using LLMs before retrieval in order to improve retrieval quality. However, these approaches do not consider the uncertainty that may arise in the answers or outputs of LLMs, which can introduce incorrect or misleading information into the retrieval process. CoopRAG addresses this limitation by guiding the LLM to identify and mask uncertain spans within the query. This prevents unreliable information from influencing retrieval results and ensures that only trustworthy parts of the reasoning chain are used for document retrieval. Furthermore, CoopRAG introduces a cooperative reasoning mechanism. After the retrieval, the LLM revisits the masked reasoning chain and fills them in, using factual evidence from the retrieved documents. This two-stage process improves not only retrieval accuracy but also reasoning robustness by grounding uncertain parts of the query in verifiable information.
>
> Regarding your second point, we have demonstrated through extensive experiments that RaLa outperforms other retrieval training objectives. Detailed results of this comparison are provided in our response to W2.
>
> ----------
>
> ### **W2) Baselines: Related to the point above, to my knowledge (correct me if I’m wrong), all baseline methods do not train their retrieval systems on the target datasets by default, as their main focus is more on inference-time algorithms and interactions between the retriever and the LM. In contrast, the proposed method trains retrievers on the target datasets. This makes me wonder whether the authors could compare model performance with methods that also train retrieval models on the target datasets.**
>
> | Models        | HotpotQA |        | MuSiQue |        | 2WikiMultihopQA |        |
> | ------------- | -------- | ------ | ------- | ------ | --------------- | ------ |
> |               | R@2      | R@5    | R@2     | R@5    | R@2             | R@5    |
> | **Without Training** |        |        |         |        |                 |        |
> | Contriever    | 57.3     | 74.8   | 32.4    | 43.5   | 55.3            | 65.3   |
> | ColBERTv2     | 64.7     | 79.3   | 37.9    | 49.2   | 59.2            | 68.2   |
> | ReSCORE       | 68.2     | 80.9   | 38.6    | 49.1   | 62.2            | 73.3   |
> | CoopRAG       | 73.9     | 86.1   | 45.9    | 60.7   | 64.6            | 75.0   |
> | **Fine-tuning**      |        |        |         |        |                 |        |
> | Contriever    | 63.4     | 80.5   | 36.8    | 44.9   | 60.6            | 68.2   |
> | ColBERTv2     | 78.2     | 89.3   | 52.6    | 68.5   | 71.6            | 82.1   |
> | ReSCORE       | 82.3     | 92.6   | 54.9    | 72.3   | 78.8            | 95.1   |
> | CoopRAG       | 88.8     | 96.8   | 59.6    | 75.7   | 80.4            | 96.6   |
>
> **Table 1.** Performance Comparison of Retrievers Before and After Fine-tuning
>
> As shown in Table 1, we compare the performance of RaLa and other retrievers before and after fine-tuning. For a fair comparison, all retrievers receive only the question as input, and question unrolling is not applied. The results show that RaLa consistently outperforms other recent retriever models across all datasets. In response to your comment, we additionally include ReSCORE [1], a recent fine-tuned retriever, for comparison. Notably, after fine-tuning, RaLa outperforms the state-of-the-art retriever ReSCORE, achieving a 7.8% higher Recall@2 on HotpotQA, which demonstrates its strong retrieval capability and adaptability.
>
> ----------
>
> ### **W3) Evaluations: Evaluations are primarily conducted on simple, short-form QA datasets (synthetic multi-hop and simple single-hop questions). These datasets are often more easily decomposed, so I was curious how the proposed "unrolling" strategies would perform on more natural, complex questions, such as those in SimpleQuestions or natural search queries (e.g., Search Arena queries). Given that intermediate reasoning chains assume knowledge-triple formats, I am afraid this method may not generalize well beyond entity-centric, factoid questions.**
>
> | Models    | SimpleQA |(Document: 47,229)      |      | FreshQA  | (Document: 11,602)     |      |      |
> | --------- | --------------------------- | ---- | ---- | -------------------------- | ---- | ---- | ---- |
> |           | EM                          | F1   |      | EM                         | F1   | Correct | Incorrect | Not Attempted |
> | HippoRAG2 | 48.2                        | 55.0 |      | 21.3                       | 29.5 | 135     | 387       | 31            |
> | HopRAG    | 50.2                        | 58.2 |      | 21.1                       | 28.7 | 147     | 361       | 45            |
> | CoopRAG   | 58.3                        | 67.6 |      | 26.6                       | 35.3 | 283     | 250       | 23            |
>
>
> **Table 2.** Performance comparison on the SimpleQA and FreshQA datasets.
>
> Both SimpleQuestions and Search Arena Queries are not compatible with the document-based QA setting and our evaluation setup. SimpleQuestions is a dataset of the KGQA task that requires reasoning over knowledge graphs rather than retrieving from text corpora. Search Arena is a human preference dataset where multiple LLM responses to a query are compared by human judgement, and ground truth answers are not available.
>
> Instead, we evaluated CoopRAG on two recent factual QA benchmarks: SimpleQA [2] and FreshQA [3]. As shown in Table 2, CoopRAG outperforms the state-of-the-art baselines HippoRAG2 and HopRAG on both datasets in terms of EM and F1. CoopRAG achieves 16.1% and 26.1% higher EMs than HopRAG on SimpleQA and FreshQA, respectively. Since FreshQA includes many sentence-level answers, we also use the ChatGPT grader from the SimpleQA paper to assess factual correctness. With this grader, CoopRAG produces 283 factually correct answers out of 553 queries, compared to only 147 by exact match, which is 92% more correct answers than HopRAG. These results demonstrate that CoopRAG generalizes well to recent and challenging QA benchmarks, showing even greater performance gains in these settings.
>
>
> ### **Reference**
> [1] Dosung Lee, Wonjun Oh, Boyoung Kim, and etc. 2025. ReSCORE: Label-free Iterative Retriever Training for Multi-hop Question Answering with Relevance-Consistency Supervision. In Proceedings of the 63rd Annual Meeting of the Association for Computational Lin- guistics (Volume 1: Long Papers), Wanxiang Che, Joyce Nabende, Ekaterina Shutova, and Mohammad Taher Pilehvar (Eds.). Association for Computational Linguistics, Vienna, Austria, 341–359. https://aclanthology.org/2025.acl-long.16/
> [2] Jason Wei, Nguyen Karina, Hyung Won Chung, and etc. 2024. Measuring short-form factuality in large language models. arXiv:2411.04368 [cs.CL] https://arxiv.org/ abs/2411.04368
> [3] Tu Vu, Mohit Iyyer, Xuezhi Wang, and etc. 2023. FreshLLMs: Refreshing Large Language Models with Search Engine Augmentation. arXiv:2310.03214 [cs.CL] https://arxiv.org/abs/2310.03214

---

> > ### Comment · Reviewer_h3Qx · 2025-08-05
> > **Thank you for your response!**
> >
> > I updated my score based on your response!

---

> ### Author Response · Authors · 2025-08-05
> **Response for h3Qx**
>
> ### **Dear Reviewer h3Qx,**
> Thank you very much for recognizing the value of our work and for raising your score. We truly appreciate your thoughtful evaluation.
>
> We hope our detailed response has fully addressed your concerns. If you have any further questions, we would be honored to address them.
>
> Thank you once again for reviewing our paper.
> Best regards,
> Authors of Paper 27544

---

### Official Review · Reviewer_pyhp · 2025-07-05

**Clarity:** 2
**Significance:** 2
**Originality:** 2
**Rating:** 3
**Confidence:** 4

**Summary:**

The paper introduces CoopRAG, a framework for RAG in QA. CoopRAG aims at addressing issues in existing RAG systems, especially in multi-hop QA setup, by fostering cooperation between the retriever and large language model. The core innovations include: 1/ decomposing the original question into sub-questions and a reasoning chain with masked uncertain elements, 2/ using the unrolled question to retrieve documents, leveraging the internal knowledge of the LLM, 3/ reranking retrieved documents by contrasting representations from different layers of the retriever.  Experimental results on three multi-hop QA datasets (HotpotQA, 2WikiMultihopQA, MuSiQue) and one QA dataset (NaturalQuestions) show that CoopRAG outperforms state-of-the-art QA methods in both retrieval and answer accuracy.

**Questions:**

see Strengths And Weaknesses

**Ethical Concerns:**

["NO or VERY MINOR ethics concerns only"]

**Final Justification:**

I thank the authors for their work and effort in addressing my concerns. I do think that the paper would benefit from another round of review given the amount of changes that are needed to incorporate the feedback. I'll keep my score to 3.

**Limitations:**

No concerns

**Paper Formatting Concerns:**

No concerns

**Quality:**

2

**Strengths And Weaknesses:**

The paper is technically sound, and the proposed framework yields promising results. However, the complexity of the approach is a key limitation that requires deeper discussion:
1) While query decomposition is a well-established method for improving RAG performance, it introduces overhead through multiple LLM calls and relies heavily on the model’s reasoning capabilities. The effectiveness of question unrolling and reasoning chain completion similarly depends on the LLM's strength. An ablation or analysis on the LLM’s impact would strengthen the claims.

2) The experimental setup could also be more rigorous. The use of historical benchmarks is a limitation and make it hard to understand the generalization capabilities of this approach. Using more recent and challenging datasets (e.g., SimpleQA, FreshLLM) would provide a more robust evaluation.

3) Comparisons with baselines should include complexity and latency metrics to better contextualize the practical trade-offs of the proposed method. Finally, I would encourage authors to add details on how they implemented the QA performance metrics.

---

> ### Author Rebuttal · Authors · 2025-07-29
>
> # Dear Reviewer pyhp,
> Thank you very much for your thoughtful and constructive feedback. We truly appreciate your time and valuable comments. We have carefully addressed each of your suggestions and will incorporate all feedback and new results into the final version. If you find our responses satisfactory, we kindly request that you reconsider the evaluation of our work and, if possible, raise the score above the middle range. Your support would be greatly appreciated. If you have any further questions or suggestions, we would be happy to provide clarification. Thank you again for your consideration.
>
> ----------
>
> ### **W1) While query decomposition is a well-established method for improving RAG performance, it introduces overhead through multiple LLM calls and relies heavily on the model’s reasoning capabilities. The effectiveness of question unrolling and reasoning chain completion similarly depends on the LLM's strength.**
> | Models                  | Retrieval |          | Reasoning |      |
> | ----------------------- | --------- | -------- | --------- | ---- |
> |                         | Reacll@2  | Reacll@2 | EM        | F1   |
> | HippoRAG2 (GPT-4o-mini) | 80.5      | 95.7     | 56.3      | 71.1 |
> | HopRAG (GPT-4o-mini)    | 81.1      | 96.0     | 62.0      | 76.1 |
> | CoopRAG (Gemma3-1B)     | 84.9      | 96.2     | 64.0      | 77.9 |
> | CoopRAG (Gemma2-2B)     | 83.7      | 96.4     | 62.4      | 76.6 |
> | CoopRAG (Llama3.3-70B)  | 86.9      | 96.6     | 64.7      | 79.0 |
>
> **Table 1.** Performance comparison of CoopRAG with varying LLM scales.
>
> To assess the robustness of CoopRAG under limited reasoning capacity, we have conducted additional experiments on the HotpotQA dataset using small-scale LLMs such as Gemma3-1B and Gemma2-2B. As shown in Table 1, CoopRAG outperforms two state-of-the-art baselines, HippoRAG2 and HopRAG, both powered by GPT-4o-mini, in terms of both retrieval and reasoning. Notably, **the performance gap for CoopRAG between using Llama3.3-70B and using the small LLMs is marginal**. These results demonstrate that CoopRAG’s performance gains are not merely due to the reasoning power of large LLMs, but stem from its uncertainty-aware query decomposition and reasoning chain completion strategies. This confirms that CoopRAG remains robust and effective even when operating with smaller language models.
>
> | Models            | Retrieval Performance |          | QA Performance |      | LLM Calls per question |          |           |
> | ----------------- | --------------------- | -------- | -------------- | ---- | ---------------------- | -------- | --------- |
> |                   | Recall@2              | Recall@5 | EM             | F1   | Preprocessing          | Retrieve | Reasoning |
> | HippoRAG2         | 80.5                  | 95.7     | 56.3           | 71.1 | 4                      | 1        | 1         |
> | HopRAG            | 81.1                  | 96.0     | 62.0           | 76.1 | 12                     | 14.96    | 1         |
> | CoopRAG (Unified) | 88.8                  | 96.8     | 63.1           | 76.6 | 0                      | 1        | 1         |
> | CoopRAG           | 88.8                  | 96.8     | 65.6           | 78.9 | 0                      | 1        | 2         |
>
> **Table 2.** Comparison of LLM Call Counts and Model Performance
>
> Furthermore, we compare the number of LLM calls required by CoopRAG and the state-of-the-art baselines, i.e.,  HippoRAG2 and HopRAG, across preprocessing, retrieval, and inference stages. As shown in Table 2, HippoRAG2 and HopRAG require 4 and 12 LLM calls per question during preprocessing, respectively, while **CoopRAG needs no preprocessing and thus requires no LLM calls at this stage**. Notably, both HopRAG and HippoRAG2 repeatedly invoke the LLM for each document to generate triples, which leads to a substantial increase in the number of LLM calls during preprocessing. During retrieval, HopRAG further incurs an average of 14.9 LLM calls per question because it uses a graph-based iterative triple extraction method. In the inference stage, CoopRAG calls the LLM twice per question. One call is for reasoning chain completion, and the other is for reasoning. Despite this, CoopRAG achieves up to 16.5% higher EM compared to the baselines. **CoopRAG (Unified), which combines the two reasoning steps into a single LLM call, also outperforms both baselines**. In summary, CoopRAG is more efficient and more effective, requiring fewer LLM calls to achieve superior performance.
>
> ----------
>
> ### **W2) The experimental setup could also be more rigorous. The use of historical benchmarks is a limitation and make it hard to understand the generalization capabilities of this approach. Using more recent and challenging datasets (e.g., SimpleQA, FreshLLM) would provide a more robust evaluation.**
>
> | Models    | SimpleQA |(Document: 47,229)      |      | FreshQA  | (Document: 11,602)     |      |      |
> | --------- | --------------------------- | ---- | ---- | -------------------------- | ---- | ---- | ---- |
> |           | EM                          | F1   |      | EM                         | F1   | Correct | Incorrect | Not Attempted |
> | HippoRAG2 | 48.2                        | 55.0 |      | 21.3                       | 29.5 | 135     | 387       | 31            |
> | HopRAG    | 50.2                        | 58.2 |      | 21.1                       | 28.7 | 147     | 361       | 45            |
> | CoopRAG   | 58.3                        | 67.6 |      | 26.6                       | 35.3 | 283     | 250       | 23            |
>
> **Table 3.** Performance comparison on the SimpleQA and FreshQA dataset.
>
> We have conducted experiments on the SimpleQA [1] and FreshQA [2] datasets, comparing CoopRAG with the two stage-of-the-art baselines. As shown in Table 3, CoopRAG outperforms the baselines in both EM and F1 scores. CoopRAG achieves 16.1% higher EM than HopRAG on SimpleQA and 26.1% higher EM on FreshQA, respectively. Since FreshQA contains many sentence-level answers, we also evaluate factual correctness using the ChatGPT grader proposed in the SimpleQA paper. Answers are graded as ‘Correct’ if the prediction fully contains the ground truth answer without contradiction, ‘Incorrect’ if there exists any contradiction (even partial or hedged), and ‘Not Attempted’ if necessary information is missing but not contradicted. For both inference and grading, we use GPT-4o-mini for all models. Importantly, while CoopRAG achieves an EM of 26.6% (147 out of 553 questions), the ChatGPT grader finds 283 answers factually correct. This highlights that EM may underestimate the actual factual correctness
>
> Compared to HopRAG, which produces only 147 factually correct answers, CoopRAG yields over 92% more factually correct responses. These results demonstrate that CoopRAG generalizes even better to recent and challenging datasets, highlighting larger performance gains on such benchmarks.
>
> ----------
>
> ### **W3) Comparisons with baselines should include complexity and latency metrics to better contextualize the practical trade-offs of the proposed method. Finally, I would encourage authors to add details on how they implemented the QA performance metrics.**
>
> | Models     | HotpotQA |      |      | MuSiQue |      |      |
> |------------|----------|------|------|---------|------|------|
> |            | Retrieval| QA   | Total| Retrieval| QA   | Total|
> | HippoRAG2  | 2.25     | 1.96 | 4.21 | 2.11    | 1.81 | 3.92 |
> | CoopRAG    | 2.04     | 1.86 | 3.90 | 1.98    | 1.65 | 3.63 |
>
> **Table 4.** Latency (seconds) of HippoRAG2 and CoopRAG on HotpotQA and MuSiQue. Lower values indicate higher efficiency.
>
> First, CoopRAG uses a retriever and reranking pipeline. The time complexity for retrieval is $O(N \cdot d)$ where $N$ is the number of documents and $d$ is the embedding dimension. For reranking, the time complexity of RaLa without our optimization as in Equation 3 of Section 3.4 is $O(L_q \cdot L_d \cdot d \cdot \vert B \vert)$, where $L_q$ is the query length, $L_d$ is the document length, $\vert B \vert$ is the bucket size. According to our optimization strategy in Equation 4 of Section 3.4, only $[CLS]$ tokens are compared per bucket, reducing the complexity to $O(L_q \cdot L_d \cdot d)$, which is asymptotically identical to that of ColBERT. In all our experiments, we use this optimized version. As shown in Table 4, CoopRAG achieves lower latency than HippoRAG2 for retrieval, reasoning, and end-to-end processing, while also delivering better retrieval and QA performances. We will provide comprehensive efficiency comparisons across all datasets in the efficiency section, Appendix D.
>
> Second, we compute all metrics following HippoRAG2 under the same evaluation setting. We report Recall@k for retrieval performance and use EM and F1 scores to evaluate QA performance. We will add a description of how we conduct experiments for both our model and the baselines, including implementation details and evaluation procedures, in Appendix A.
>
> ----------
>
> ### **Reference**
> [1] Jason Wei, Nguyen Karina, Hyung Won Chung, and etc. 2024. Measuring short-form factuality in large language models. arXiv:2411.04368 [cs.CL] https://arxiv.org/ abs/2411.04368
> [2] Tu Vu, Mohit Iyyer, Xuezhi Wang, and etc. 2023. FreshLLMs: Refreshing Large Language Models with Search Engine Augmentation. arXiv:2310.03214 [cs.CL] https://arxiv.org/abs/2310.03214

---

> ### Author Response · Authors · 2025-08-07
>
> ### **Dear Reveiwer pyhp,**
> We sincerely appreciate you for recognizing the contributions of our research and for providing valuable feedback. We are very grateful to you for high quality reviews and constructive comments. We tried to fully respond to every single comment and suggestion. We will incorporate your feedback and the additional experimental results into the final version to enhance the completeness of our paper.
>
> If you find our responses satisfactory, we kindly request that you reconsider the evaluation of our work. Should you have any additional suggestions or comments, we are always more than willing to provide clarifications to address them. Many thanks!
>
> Thank you once again for your thoughtful comments and your time

---

### Official Review · Reviewer_S8QW · 2025-07-23

**Clarity:** 3
**Significance:** 3
**Originality:** 3
**Rating:** 4
**Confidence:** 4

**Summary:**

The paper proposes a retrieval-augmented generation approach called CoopRAG for multi-hop QA and single hop QA tasks.

The proposed approach consists of multiple modules in sequence: (a) retriever (b) re-ranker, and (c) (generative) language model. The main novelty is in the design and training methodology of the individual components and jointly. Given a query, first the language model uses its knowledge to decompose a question into multiple sub-questions, along with reasoning knowledge triples required to answer the question. To emphasize the LM’s uncertainty in the reasoning chain, it is allowed to output masked tokens with placeholder tokens. Given this information, the retriever selects the top-n documents from candidate pool which is fed to re-ranker module to further select the most relevant top-k documents (K < N). The re-ranker is trained using a contrastive layers approach while the retriever training loss is modified to give importance to difficult examples. The answer is generated given the re-ranked documents along with the incomplete steps of the reasoning chain.

Experimental results on three multi-hop and single-hop datasets are state-of-the-art. A set of ablation studies highlights the importance individual design decisions such as training methodology and LM prompt design for the reasoning chain.

**Questions:**

Please see the Weakness section above.

**Ethical Concerns:**

["NO or VERY MINOR ethics concerns only"]

**Limitations:**

yes

**Quality:**

3

**Strengths And Weaknesses:**

Strengths:
* The proposed system consists of well designed multiple sub-components. The LM is encouraged to expand the query by asking sub-questions, provide a reasoning chain, and using accurate set of retrieved documents to first complete the reasoning trace and answer the query. A couple of novel ideas are presented to improve the retrieved set of documents - First, by contrastively using the transformer layers in ranking and giving more weight to the difficult examples during model training process. These design choices lead to accuracy improvements in end evaluations.
* Good performance results: The effectiveness of the proposed system design of CoopRAG is shown in obtaining new state-of-the-art results across four hard datasets on both their retrieval and end QA task, which is quite significant. They have compared the method against a range of recent baselines.


Weaknesses:
* My main point of concern is that the method is not really tested at scale. As shown in Table 1 (Sec 4.1), the candidate size of the evidence pool is quite smaller (on the range of 10000 documents). To really understand how these methods would perform in real-world QA tasks, ideally, the candidate size of Wikipedia should be considered and evaluated against. On smaller candidate size, its easy to get a very high retrieval recall as there aren’t many strong distracting negatives.
* More evaluation datasets needed: The datasets such as HotpotQA and Natural Questions are now somewhat old and so there are chances that their reference answers won’t be conflicting when provided with updated knowledge bases such as Wikipedia. It would be nice to have some results on more recent datasets such as SimpleQA.

---

> ### Author Rebuttal · Authors · 2025-07-29
>
> # **Dear Reveiwer S8QW**
> Thank you very much for your thoughtful and constructive feedback. We truly appreciate your time and valuable comments. We have carefully addressed each of your suggestions, and we will incorporate all feedback and new results into the final version. If you find our responses satisfactory, we kindly request that you reconsider the evaluation of our work and, if possible, raise the score above the middle range. Your support would be greatly appreciated. If you have any further questions or suggestions, we would be happy to provide clarification. Thank you again for your consideration.
>
> ----------------
>
> ###  **W1) My main point of concern is that the method is not really tested at scale. As shown in Table 1 (Sec 4.1), the candidate size of the evidence pool is quite smaller (on the range of 10000 documents). To really understand how these methods would perform in real-world QA tasks, ideally, the candidate size of Wikipedia should be considered and evaluated against. On smaller candidate size, its easy to get a very high retrieval recall as there aren’t many strong distracting negatives.**
>
> | Models                | Candidate |  Size |       |       |       |       |
> | --------------------- | -------------- | ----- | ----- | ----- | ----- | ----- |
> |                       | 10000          | 20000 | 30000 | 40000 | 50000 | 60000 |
> | HippoRAG2             | 95.7           | 86.5  | 78.2  | 56.3  | 41.7  | 33.2  |
> | HopRAG                | 96.0           | 85.6  | 77.1  | 58.8  | 43.4  | 38.9  |
> | CoopRAG               | 96.8           | 88.6  | 81.9  | 68.7  | 59.6  | 56.8  |
> | Performance  Gain (%) | 0.83           | 2.43  | 4.73  | 16.84 | 37.33 | 46.02 |
>
> **Table1.** Retrieval performance  of CoopRAG and main baselines at different Wikipedia candidate scales.
>
> | Models    | Candidate |  Size |       |       |       |       |
> | --------- | -------------- | ----- | ----- | ----- | ----- | ----- |
> |           | 10000          | 20000 | 30000 | 40000 | 50000 | 60000 |
> | HippoRAG2 | 2.25           | 2.72  | 2.88  | 3.56  | 3.89  | 4.32  |
> | HopRAG    | 13.3           | 15.3  | 16.0  | 20.2  | 24.3  | 29.6  |
> | CoopRAG   | 2.01           | 2.28  | 2.53  | 2.79  | 3.05  | 3.26  |
>
> **Table2.**  Latency  of CoopRAG and main baselines at different Wikipedia candidate scales.
>
> We assess the scalability of CoopRAG by varying the candidate size in Wikipedia. Table 1 shows that as the number of candidates increases, CoopRAG exhibits much less performance degradation than the state-of-the-art baselines HippoRAG2 and HopRAG. With 10,000 documents, CoopRAG achieves a 0.83% gain over HopRAG, which increases to 46.02% at 60,000 documents. Furthermore, we also measure the retrieval time as the number of candidates increases, as shown in Table 2. CoopRAG consistently demonstrates faster retrieval times than all baselines, regardless of candidate size.Thus, CoopRAG remains more robust than other baselines even in large candidate settings.
>
> ----------------
>
> ### **W2) More evaluation datasets needed: The datasets such as HotpotQA and Natural Questions are now somewhat old and so there are chances that their reference answers won’t be conflicting when provided with updated knowledge bases such as Wikipedia. It would be nice to have some results on more recent datasets such as SimpleQA.**
> | Models    | SimpleQA  |      |
> | --------- | -------------------------- | ---- |
> |           | EM                         | F1   |
> | HippoRAG2 | 48.2                       | 55.0 |
> | HopRAG    |  50.2                       | 58.2 |
> | CoopRAG   | **58.3**                       | **67.6** |
>
> **Table 3.** Performance comparison on the SimpleQA dataset, where the number of documents is 47,229.
>
> We have conducted an additional experiment comparing CoopRAG with the two recent baselines, HippoRAG2 and HopRAG, on the SimpleQA dataset. As shown in Table 3, CoopRAG outperforms the baselines in both EM and F1 scores, achieving 16.1% higher EM than HopRAG. These results demonstrate that CoopRAG generalizes well to recent and challenging datasets, thereby supporting its robustness beyond historical benchmarks.

---

> ### Author Response · Authors · 2025-08-07
>
> ### **Dear Reveiwer S8QW,**
> We sincerely appreciate you for recognizing the contributions of our research and for providing valuable feedback. We are very grateful to you for high quality reviews and constructive comments. We tried to fully respond to every single comment and suggestion. We will incorporate your feedback and the additional experimental results into the final version to enhance the completeness of our paper.
>
> If you find our responses satisfactory, we kindly request that you reconsider the evaluation of our work. Should you have any additional suggestions or comments, we are always more than willing to provide clarifications to address them. Many thanks!
>
> Thank you once again for your thoughtful comments and your time.

---

### Note · Authors · 2025-08-12

We sincerely thank all reviewers for their valuable feedback. Among NeurIPS 2024 papers on similar topics, RankRAG [1], GR [2], and CoKe [3] were accepted with high scores as spotlights, and G-Retrieval [4] with a score above borderline. This demonstrates the active growth of this research area and supports that CoopRAG makes contributions of comparable significance.

Reviewers have acknowledged the following strengths of CoopRAG: it presents a well-designed architecture to enhance retrieval and reasoning in QA tasks. RaLa contrasts higher and lower transformer layers during training and assigns greater weight to difficult examples, representing a distinctive contribution not seen in prior work. These design choices lead to substantial accuracy gains, achieving SOTA results across four challenging QA benchmarks, while outperforming various recent baselines. In addition to its technical soundness and clarity, CoopRAG’s methodological innovations contribute to its robust and consistent performance improvements.

During the rebuttal period, we have addressed all reviewers' concerns: (i) for **Reviewer S8QW**, we ran a scalability experiment where CoopRAG shows far less performance drop than SOTA and up to 46.02% gain with lower latency; evaluated on SimpleQA, where EM increases by 16.1%, (ii) for **Reviewer pyhp**, we tested small-scale LLMs, achieving minimal drop while surpassing SOTA with fewer calls; evaluated on SimpleQA and FreshQA, where EM increases by 16.1% and 26.1% respectively with higher factual correctness; conducted complexity analysis, confirming lower latency, (iii) for **Reviewer h3Qx**, we compared RaLa to fine-tuned retrievers (e.g., ReSCORE), showing higher Recall; evaluated on SimpleQA and FreshQA, where EM increases by 16.1% and 26.1% with over 92% factually correct answers, (iv) for **Reviewer aR4a**, we referenced four case studies in Appendix G; performed error analysis to categorize errors. These collectively address all concerns through additional quantitative and qualitative analyses, demonstrating our contribution.

[1] Y. Yu et al. RankRAG: Unifying Context Ranking with Retrieval-Augmented Generation in LLMs.
[2] Y. Tang et al. Generative Retrieval Meets Multi-Graded Relevance.
[3] J. Dong et al. Cost-efficient Knowledge-based Question Answering with Large Language Models.
[4] X. He et al. G-Retriever: Retrieval-Augmented Generation for Textual Graph Understanding and Question Answering.

---

### Decision · Program_Chairs · 2025-09-17

**Decision:**

Accept (poster)

**Comment:**

- This paper presents Cooperative Retrieval-Augmented Generation for Question Answering, where a retriever and an LLM work cooperatively by exchanging informative knowledge, and the earlier and later layers of the retriever model work cooperatively to rank the retrieved documents relevant to a given query
- The paper proposes some interesting ideas about contrasting LLM layers for retriever training, has solid and promising results. There were several concerns (such as not evaluated at scale, missing more recent eval benchmarks), but they have been largely addressed after rebuttal. The remaining holdback is around limited novelty and whether the authors will be able to incorporate all the feedback into paper revision.
- Overall, this looks like a borderline accept to me